# Overexpression of Claspin and Timeless protects cancer cells from replication stress in a checkpoint-independent manner

Julien N. Bianco[1,7], Valérie Bergoglio[2], Yea-Lih Lin[1], Marie-Jeanne Pillaire[2], Anne-Lyne Schmitz[1], Julia Gilhodes[3], Amelie Lusque[3], Julien Mazières[4], Magali Lacroix-Triki[5], Theodoros I. Roumeliotis[6], Jyoti Choudhary[6], Jérôme Moreaux [1], Jean-Sébastien Hoffmann [2], Hélène Tourrière[1] & Philippe Pasero[1]

Oncogene-induced replication stress (RS) promotes cancer development but also impedes tumor growth by activating anti-cancer barriers. To determine how cancer cells adapt to RS, we have monitored the expression of different components of the ATR-CHK1 pathway in primary tumor samples. We show that unlike upstream components of the pathway, the checkpoint mediators Claspin and Timeless are overexpressed in a coordinated manner. Remarkably, reducing the levels of Claspin and Timeless in HCT116 cells to pretumoral levels impeded fork progression without affecting checkpoint signaling. These data indicate that high level of Claspin and Timeless increase RS tolerance by protecting replication forks in cancer cells. Moreover, we report that primary fibroblasts adapt to oncogene-induced RS by spontaneously overexpressing Claspin and Timeless, independently of ATR signaling. Altogether, these data indicate that enhanced levels of Claspin and Timeless represent a gain of function that protects cancer cells from of oncogene-induced RS in a checkpoint-independent manner.

[1] Institut de Génétique Humaine, CNRS, Université de Montpellier, Equipe Labellisée Ligue Contre le Cancer, 34396 Montpellier, France. [2] Cancer Research Center of Toulouse, INSERM U1037, CNRS ERL5294, University of Toulouse 3, 31037 Toulouse, France. [3] Clinical trials Office - Biostatistics Unit, Institute Claudius Regaud, Institute Universitaire du Cancer Toulouse-Oncopole (IUCT-O), 31100 Toulouse, France. [4] Thoracic Oncology Department, Toulouse University Hospital, University Paul Sabatier, 31062 Toulouse, France. [5] Gustave Roussy Cancer Campus, 94805 Villejuif, France. [6] The Institute of Cancer Research, London SW3 6JB, UK. [7] Present address: Cologne Excellence Cluster for Cellular Stress Responses in Ageing-Associated Diseases (CECAD), University of Cologne, Joseph-Stelzmann-Str. 26, 50931 Cologne, Germany. These authors contributed equally: Hélène Tourrière, Philippe Pasero. Correspondence and requests for materials should be addressed to H.T. (email: helene.tourriere@igh.cnrs.fr) or to P.P. (email: philippe.pasero@igh.cnrs.fr)

Genomic instability is a cancer hallmark that is detected at early stages of tumorigenesis and is generally considered as a driving force of cancer development[1]. A growing body of evidence indicates that DNA damage arises as a consequence of oncogene-induced replication stress (RS)[2–4]. RS refers to a variety of events of endogenous or exogenous origin that interfere with the progression of DNA replication forks[5,6]. In cancer cells, RS is caused by the aberrant activation of oncogenes, which may either increase conflicts between replication and transcription or uncouple DNA synthesis from nucleotide metabolism[4,7].

RS activates a surveillance pathway known as the replication checkpoint[8]. In this pathway, the ATR kinase is recruited to stressed forks by the accumulation of replication protein A (RPA)-coated single-stranded DNA and is activated by TopBP1, a factor loaded at single-stranded/double-stranded DNA junctions by the 9-1-1 complex (RAD1, RAD9, and HUS1) and its clamp loader, RFC[RAD17][8]. Once activated, ATR phosphorylates the effector kinase CHK1 on Ser317 and Ser345 to amplify the checkpoint signal. This process is mediated by Claspin, Timeless, and Tipin, which form a complex at replication forks and act as mediators for CHK1 activation[9–11]. Once activated, the ATR-CHK1 pathway acts in many ways to coordinate fork repair processes, prevent premature entry into mitosis and allow the completion of DNA replication[8].

Oncogene-induced RS is a double-edged sword. Although it contributes to cancer development by promoting genomic instability, it slows down cell proliferation and activates anticancer barriers leading to apoptosis or senescence[12–15]. To proliferate, cancer cells must therefore bypass these barriers, while avoiding severe replicative defects that are incompatible with cell survival. It is generally believed that cells adapt to oncogene-induced RS by modulating the intensity of the ATR-CHK1 checkpoint response[16–18]. Indeed, ATR and CHK1 haploinsufficiencies enhance oncogene-induced tumor formation[19,20], but a more severe depletion of ATR is synthetic lethal with oncogene overexpression[19,21]. Along the same line, a mild overexpression of CHK1 in mouse by addition of an extra-copy of the CHK1 gene decreases oncogene-induced RS and promotes tumor growth[22]. Collectively, these data indicate the ATR-CHK1 pathway has both protumoral and antitumoral activities depending on the cellular context[16,18]. Understanding how cancer cells control this balance represents therefore a major challenge in cancer biology.

Owing to their central position in the ATR-CHK1 pathway and their fork association, Claspin, Timeless, and its partner Tipin are ideally placed to fine tune the cellular response to oncogene-induced RS. These factors are upregulated in many different cancers and their increased expression is associated with bad prognosis[23–29]. Overexpression of Claspin is also a marker of radioresistance in metastasis lung cancer[30] and Timeless is a candidate molecular marker for predicting the response of ER α-positive postmenopausal breast cancer to Tamoxifen[31]. However, the mechanism by which Claspin, Timeless, and Tipin promote cancer progression is currently unclear.

Besides their role in the ATR-CHK1 pathway[11,32], Claspin, Timeless, and Tipin also play a structural role at replication forks that is independent of their checkpoint function[33–36]. Indeed, these three proteins form a complex at replication forks called the fork protection complex (FPC), which is conserved from yeast to vertebrates[32]. Mrc1, Tof1, and Csm3, the budding yeast homologs of Claspin, Timeless, and Tipin, interact with DNA polymerase ε and the CMG helicase on the leading strand synthesis[37,38] and are required for normal fork progression in a checkpoint-independent manner[39,40]. In vertebrates, Claspin is a DNA binding protein that associates with branched structures in a highly specific and strong manner and interacts with numerous components of the replication machinery, namely MCM proteins, DNA polymerases (pol) α, δ, ε, CDC7 kinase, and CDC45[32,34]. Timeless and Tipin interact with the replicative helicase components MCM2-7 and CDC45, and with replicative polymerases Pol ε and Pol δ, increasing their processivity in vitro[32]. Since replication defect in Timeless-depleted cells is synthetic with ATR depletion[35], Timeless could coordinate enzymatic activities at the fork, independently of ATR.

Here, we have investigated the mechanism by which cancer cells adapt to oncogene-induced RS by modulating components of the ATR-CHK1 pathway. We have analyzed the expression of key components of this pathway in primary lung, breast, and colon cancer samples and in a variety of cancer cell lines. We found that unlike the checkpoint sensors ATR, RAD9, and RAD17, the downstream components of the ATR pathway Claspin, Timeless, and CHK1 show a correlated overexpression in cancer cells. To characterize the protumoral effect of this increased expression, we have reduced the excess of Claspin and Timeless in HCT116 cells under conditions that do not prevent checkpoint signaling. We show that this depletion reduces fork speed, increases fork stalling and leads to accumulation of γ-H2AX, in a checkpoint-independent manner. Moreover, primary fibroblasts escaping oncogene-induced senescence overexpress Claspin and Timeless. Altogether, these data indicate that cancer cells adapt to RS by overexpressing Claspin and Timeless, independently of ATR signaling.

## Results

**Components of the ATR pathway are overexpressed in cancers.** The ATR-CHK1 pathway controls tumor progression in a dosage-dependent manner[16], but the mechanism by which tumor cells modulate this pathway to adapt to oncogene-induced RS is currently unclear. To address this question, we have analyzed the expression of the checkpoint sensors RAD17, RAD9, and ATR, the mediators Claspin and Timeless and the effector CHK1 (Fig. 1a) in 93 primary non–small-cell lung cancers (NSCLC)[27], 74 primary colorectal carcinomas[41], and 206 primary breast cancers[42]. For each gene, mRNA levels were determined by a quantitative real-time polymerase chain reaction (qRT-PCR) and were expressed relative to normal adjacent tissue (lung and colorectal cancers) or to a pool of healthy tissues (breast cancer). This analysis revealed that Claspin, Timeless, and CHK1 were overexpressed in these three cohorts (Fig. 1b; $T/N > 1$) in a correlated manner, with a Spearman coefficient ranging from 0.56 to 0.80 (Fig. 1c). In contrast, the expression of the checkpoint sensors ATR, RAD9, and RAD17 was only modestly increased in cancer samples relative to normal tissues (Fig. 1b), and did not correlate with the downstream components of the pathway (Fig. 1c). A clustered expression of Claspin, Timeless, and CHK1 was also observed when a larger number of DNA damage response (DDR) genes were analyzed (Supplementary Fig. 1c–h), supporting the view that the downstream components of the ATR-CHK1 pathway are co-regulated.

In lung cancer, the expression of Claspin, Timeless, and CHK1 correlated also to some extent with PCNA, but this correlation was not observed in colon and breast cancers (Fig. 1c). In addition, PCNA mRNA levels were only moderately increased (1.4-fold) in lung cancer compared to those of Claspin and CHK1 (4.5- and 4.4-fold, respectively; Fig. 1b), indicating that the increased expression of Claspin, Timeless, and CHK1 in cancer cells does not simply reflect increased proliferation. Altogether, these data indicate that the upstream and downstream components of the ATR-CHK1 checkpoint are differentially regulated in primary cancer cells.

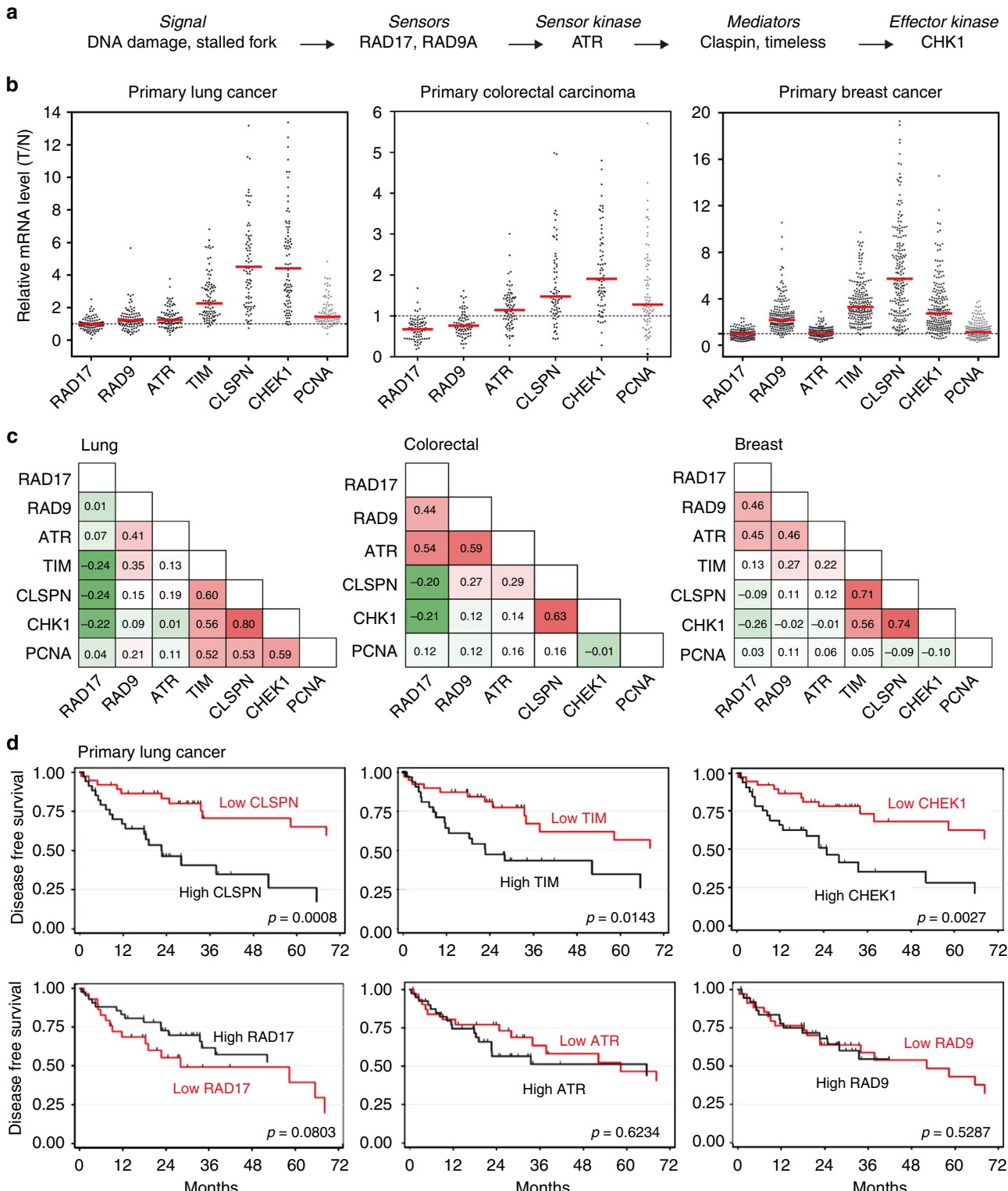

**Fig. 1** Claspin, Timeless, and CHK1 mRNA are overexpressed in primary colorectal, lung, and breast cancers. **a** Schematic representation of the ATR pathway showing the six checkpoint factors analyzed in this study. **b** Relative mRNA levels of ATR, RAD9, RAD17, Claspin, Timeless, CHK1, and PCNA in 93 lung tumors, 74 primary colorectal carcinomas, and 206 breast cancer samples. mRNA levels were determined by quantitative RT-PCR and are expressed as the ratio of tumor (T) to normal tissues (N). **c** Nonparametric Spearman correlation analysis of the expression of checkpoint genes and proliferation marker (PCNA) in lung, colorectal, and breast cancer samples. **d** High expression of Claspin, Timeless, and CHK1 is associated to bad prognosis in NSCLC patients. Disease-free survival (DFS) according to gene level expression: Claspin, Timeless, CHK1, RAD17, ATR, and RAD9. Totally, 72 patients were analyzed

**Claspin and Timeless are prognostic biomarkers of NSCLC**. To determine whether the overexpression of the downstream components of the ATR-CHK1 pathway could have an impact on cancer development, we focused on a cohort of low grade (IA, IB, IIA, and IIB) NSCLCs patients who received no adjuvant treatment and asked whether the expression level of components of the ATR-CHK1 pathway determined at diagnosis had an impact on their disease-free survival (DFS) over a 72 months period. For each of these genes, patients were separated in two groups depending on the level of the corresponding mRNAs. Remarkably, the group of patients with tumors overexpressing Claspin, Timeless, or CHK1 showed a marked decrease in DFS compared to low-expression group (Fig. 1d). In contrast, the effect of RAD9, RAD17, and ATR levels was not significant (Fig. 1d). Together, these data indicate Claspin, Timeless, and CHK1 are frequently overexpressed in primary cancers independently of upstream components of the ATR-CHK1 pathway. In lung cancer, this increased expression was associated with a reduced DFS, suggesting that it could be used as a prognostic biomarker for early NSCLC. Antibodies against Claspin and Timeless have been successfully used on paraffin-embedded tumor samples[25,26], which was confirmed here for Timeless with breast cancer samples (Supplementary Fig. 1a, b).

**Claspin and Timeless are overexpressed in cancer cell lines**. To determine how Claspin, Timeless, and CHK1 could promote tolerance to RS independently of ATR signaling, we first checked their protein levels in cancer cell lines and immortalized primary cells. Immunoblots confirmed that unlike ATR and RAD17, levels of Claspin, Timeless, and CHK1 were highly increased in transformed cells (U2OS, HeLa, and HCT116) compared to immortalized primary fibroblasts (IMR90, BJ), immortalized primary epithelial cells (RPE-1 and MCF10A) (Fig. 2a, b). Chromatin fractionation experiments also revealed that the amount of chromatin-bound Claspin and Timeless was proportionally increased in HCT116 cells (Supplementary Fig. 2), suggesting that a large fraction of overexpressed proteins was present on chromatin and could therefore play a biological role.

Next, we asked whether protein levels of Claspin, Timeless, and CHK1 are correlated in cancer cell lines. To this end, we monitored the relative abundance of 18 checkpoint proteins in the proteomic landscapes of 50 colorectal cancer cell lines[43]. This analysis confirmed that the abundance of Claspin, Timeless, and CHK1 is highly correlated and that these proteins form a cluster with BRCA1, TopBP1, and Tipin that is distinct from ATR, ATRIP, RAD9A, HUS1, and RAD1 (Fig. 2c). This is reminiscent of the correlation observed at the mRNA level in cancer patients (Supplementary Fig. 1d, f, h) and supports the view that Claspin, Timeless, and CHK1 are part of a functional module whose function is distinct from the ATR signaling pathway.

**Excess of Claspin is dispensable for CHK1 activation**. CHK1 plays pleiotropic roles in the cell and is essential for viability[44–46]. We, therefore, focused subsequent analyses on the function of Claspin and Timeless, using HCT116 colon cancer cells as an experimental model. To evaluate the functional significance of the overexpression of Claspin and Timeless in HCT116 cells, we first expressed shRNAs against these factors and observed a reduction in cell proliferation, growth on soft agar (Fig. 3a–c) and colony formation (Supplementary Fig. 3e) relative to control cells. Since the best-characterized function of Claspin and Timeless is to mediate checkpoint activation in S phase[9,10], we next monitored the ability of sh-CLSPN and sh-TIM cells to promote CHK1 phosphorylation in response to nucleotide depletion by hydroxyurea (HU). Remarkably, CHK1 was efficiently phosphorylated

on Ser 317 in sh-CLSPN and sh-TIM cells (Fig. 3d) and the CHK1 target CDC25A was rapidly degraded in a CHK1-dependent manner in HU-treated sh-CLSPN and sh-TIM cells (Fig. 3e). Similar results were also obtained with a lower dose of HU (100 μM; Supplementary Fig. 3a) or after a transient transfection with siRNAs against Claspin, Timeless, or both proteins (Supplementary Fig. 3b). These data indicate that the residual levels of Claspin and Timeless in sh-CLSPN and sh-TIM cells are sufficient for the timely activation of the ATR-CHK1 pathway in response to RS. Efficient CHK1 activation was also observed in U2OS and MCF7 cells transfected with siRNAs against Claspin and Timeless (Supplementary Fig. 3c, d), under conditions that interfere with the growth of MCF7 cells (Supplementary Fig. 3f). Altogether, these data indicate that the overexpression of Claspin and Timeless in cancer cell lines promotes cell growth independently of their checkpoint function.

**High levels of Claspin and Timeless promote fork progression**. In unchallenged growth conditions, the reduction of Claspin and Timeless levels induced the formation of γ-H2AX in HCT116 cells (Fig. 4c) and in U2OS and MCF7 cells (Supplementary Fig. 3c–d), which may reflect the induction of spontaneous RS. Since H2AX can also be phosphorylated by ATM and DNA-PK outside of S phase[47], ongoing DNA replication was labeled with EdU to specifically monitor γ-H2AX levels in S-phase cells. Flow cytometry analyses revealed that 60% and 69% of EdU-positive cells were enriched in γ-H2AX upon reduction of Claspin or Timeless levels, respectively, whereas it is only the case in 22% of untreated HCT116 cells (Supplementary Fig. 4a). Immunofluorescence experiments confirmed that γ-H2AX levels in EdU-positive cells were also significantly higher in sh-CLSPN and sh-TIM cells relative to control cells (Fig. 4a, b). These data indicate that the shRNA-mediated reduction Claspin and Timeless levels in HCT116 cells increases spontaneous RS.

Next, we used DNA combing to monitor the impact of Claspin and Timeless levels on the progression of individual replication forks. To this end, sh-Ctrl, sh-CLSPN, and sh-TIM HCT116 cells were labeled for 10 min with IdU and for 20 min with CldU. DNA fibers were stretched on silanized coverslips and the length of CldU tracks was measured as described previously[48]. CldU tracks were significantly shorter in sh-CLSPN and sh-TIM cells than in sh-Ctrl cells (16.6 and 18.4 kb, respectively, vs. 22.1 kb in control cells, $p < 0.001$), indicating that replication forks are ~25% slower upon reduction of Claspin or Timeless levels (Fig. 4d). This slow fork progression was also confirmed with a related DNA fiber assay called DNA fiber spreading[49], using inducible shRNAs (Supplementary Fig. 4b) or siRNAs (Supplementary Fig. 4c) against Claspin and Timeless or both (Supplementary Fig. 4c). To determine whether this slow fork phenotype is due to an increased rate of fork stalling, we measured the level of asymmetry between the distance covered by sister replication forks, as described earlier[50]. This analysis revealed a significant increase in fork asymmetry upon reduction of Claspin or Timeless levels in HCT116 cells (Supplementary Fig. 4d). Together, these data indicate that high levels of Claspin and Timeless in HC116 cells protect them from endogenous RS, independently of their ability to activate CHK1.

**Claspin and Timeless promote genome stability in S phase**. The budding yeast homologs of Claspin and Timeless, namely Mrc1 and Tof1, promote replication fork progression through sequences that are intrinsically difficult to replicate, in a checkpoint-independent manner[39,51,52]. Common fragile sites (CFSs) are regions of the human that are replicated in late S phase and break more frequently under RS conditions[53,54]. To

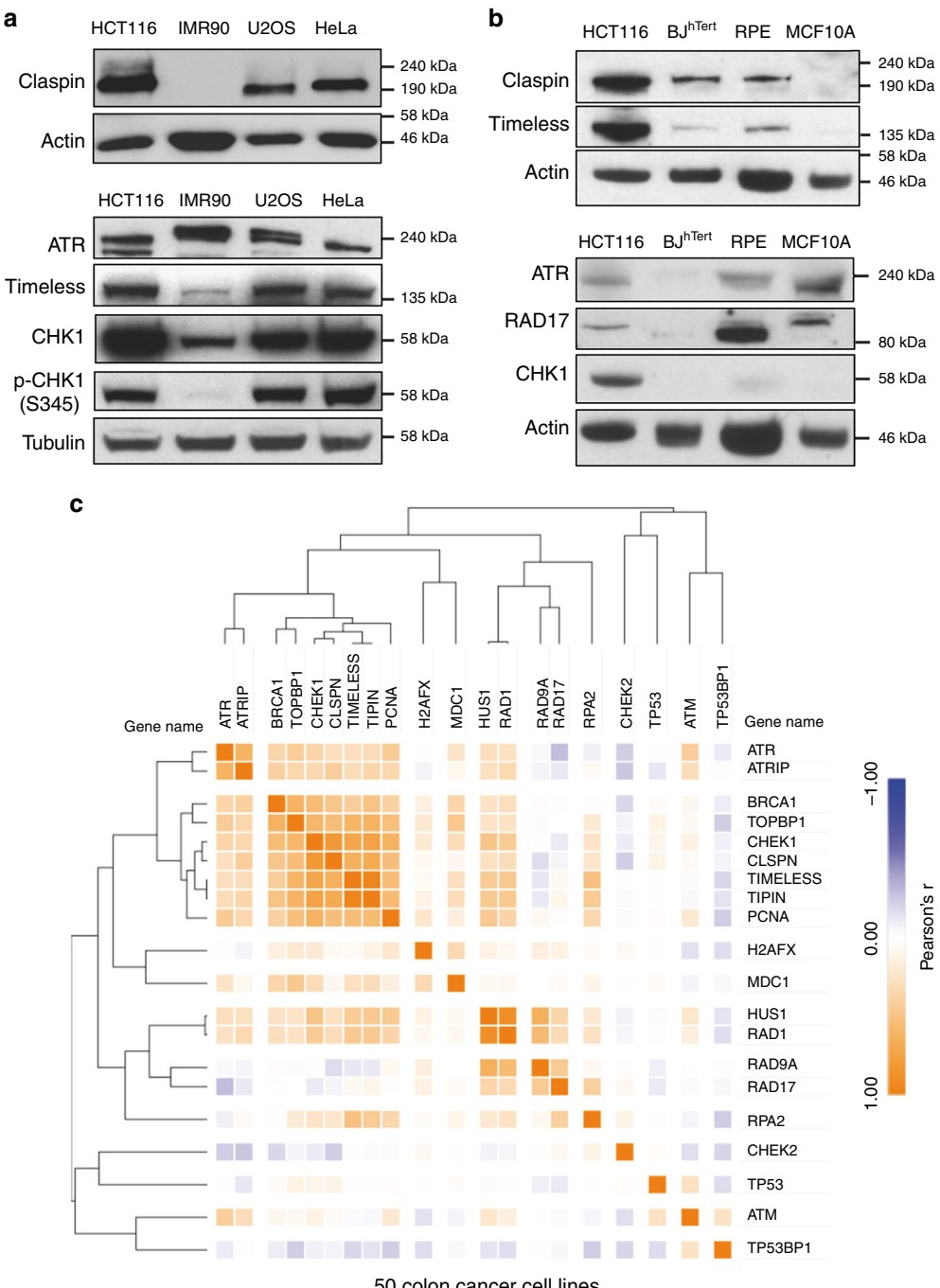

**Fig. 2** Coordinated upregulation of Claspin, Timeless, and CHK1 protein levels in cancer cell lines. **a** Western blot analysis of the levels of Claspin, Timeless, CHK1, and ATR in human cancer cell lines (U2OS, HeLa, and HCT116) and in immortalized primary fibroblasts (IMR90). **b** Claspin, Timeless, CHK1, RAD17, and ATR in colorectal cancer HCT116 cells, immortalized primary fibroblasts (BJ) and immortalized epithelial cells (RPE-1 and MCF10A). **c** Correlation analysis of the abundance of DDR proteins in 50 colorectal cancer cell lines

determine whether late-replicating regions are particularly difficult to replicate in the absence of Claspin and Timeless, sh-Ctrl, sh-CLSPN, and sh-TIM HCT116 cells were sorted by flow cytometry and were collected in $G_1$, early S, and late S phase (Fig. 4e). Immunoblotting of γ-H2AX in each subpopulation confirmed that γ-H2AX levels were globally higher during S phase in sh-CLSPN and sh-TIM cells compared to control cells and revealed that γ-H2AX was preferentially enriched at late-replicating regions in the absence of Timeless (Fig. 4f). We also monitored replication fork progression in the same samples by DNA combing. In sh-CLSPN cells, we observed an overall reduction of

fork speed relative to control cells, regardless of the S-phase stage (Fig. 4g). In sh-TIM cells, this slowdown of the forks was only detected in late S phase, which is consistent with their increased level of γ-H2AX (Fig. 4f). Since CFSs replicate late during S phase, we next asked whether these loci break more frequently in the absence of Claspin and Timeless. To this end, we have analyzed copy-number variations (CNVs) and loss of heterozygosity in these cells by single-nucleotide polymorphism comparative genomic hybridization (SNP-CGH). This analysis revealed 33 recurrent rearrangements in sh-Ctrl cells (Supplementary Fig. 4e), which are typical of HCT116 cells[55]. In addition, 14 specific

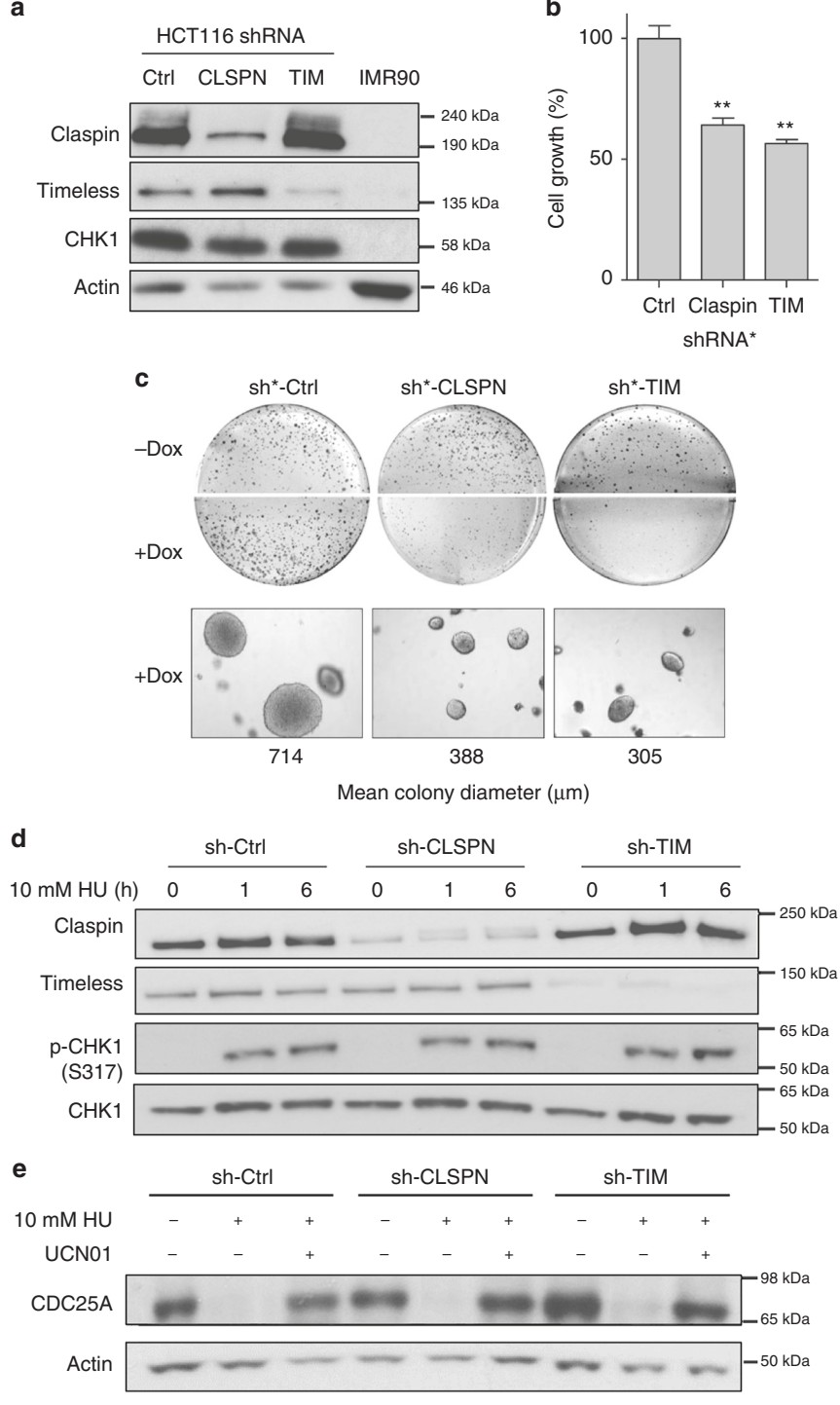

**Fig. 3** The overexpression of Claspin and Timeless promotes cell growth in a checkpoint-independent manner. **a** Levels of Claspin, Timeless, and CHK1 in immortalized IMR90 fibroblasts and in HCT116 cells transduced with a control shRNA (sh-Ctrl) or with shRNAs against Claspin (sh-CLSPN) and Timeless (sh-TIM). **b** Growth of HCT116 transduced with doxycycline-inducible shRNAs against Claspin (sh*-CLSPN) or Timeless (sh*-TIM) or with a control shRNA (sh*-Ctrl). Cell growth was determined 48 h after addition of doxycyclin. **p < 0.05. **c** Anchorage-independent growth assay. HCT116 cells transduced with sh*-Ctrl, sh*-CLSPN and sh*-TIM inducible shRNAs were grown 2 weeks in soft agar medium (0.7%) in the absence (−Dox) or the presence (+Dox) of Doxycyclin. Colonies were visualized by microscopy and mean colony size was determined with MetaMorph. Representative images are shown. **d** Western blot analysis of CHK1 phosphorylation (p-CHK1 and S317) in asynchronous sh-Ctrl, sh-CLSPN, and sh-TIM HCT116 cells treated for 0, 1, or 6 h with 10 mM HU. **e** Checkpoint-dependent degradation of CDC25A in sh-Ctrl, sh-CLSPN, and sh-TIM HCT116 cells treated for 0 or 6 h with 10 mM HU and 300 nM of the CHK1 inhibitor UCN01

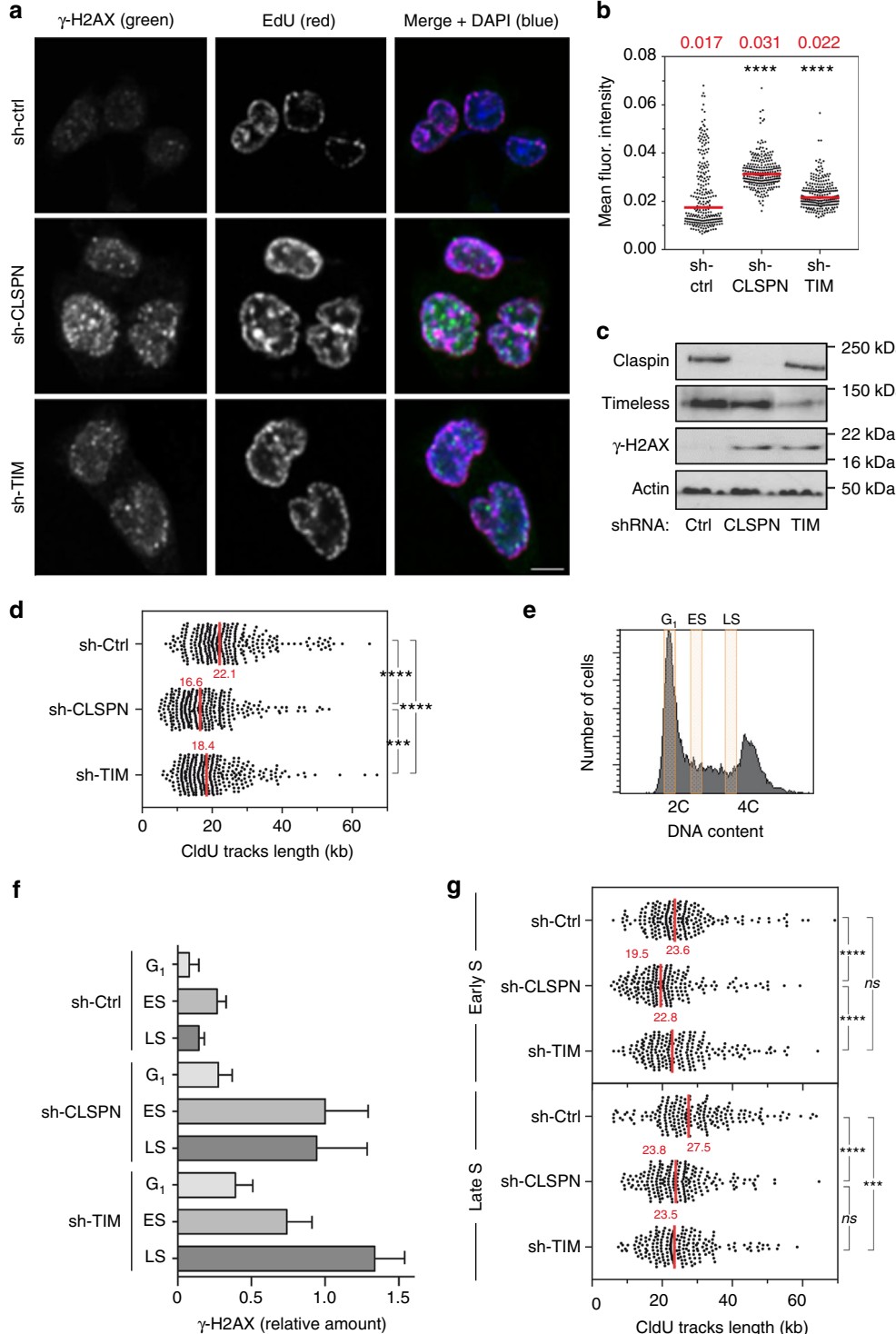

rearrangements were detected in sh-CLSPN cells and 22 in sh-TIM cells, almost half of which colocalized with CFSs (Supplementary Fig. 4e, f). Collectively, these data indicate that increased levels of Claspin and Timeless promote replication fork progression and prevent genomic instability in regions of the genome that are intrinsically difficult to replicate, such as CFSs.

**Claspin and Timeless protect cells from oncogenic RS.** Spontaneous DNA repair foci are frequently detected at early stages of tumorigenesis as a consequence of oncogene-induced

RS[3,7,12,13,56,57], but generally disappear at later stages, which suggests that cancer cells adapt to RS. To determine whether the overexpression of Claspin and Timeless helps cells tolerate oncogene-induced RS, an oncogenic form of Ras (Ras$^{V12}$) was expressed in immortalized BJ-hTERT fibroblasts with a doxycyclin-inducible promoter. As expected, Ras$^{V12}$ increased the levels of phospho-CHK1 and γH2AX (Supplementary Fig. 5a) and induced replication fork progression defects (Supplementary Fig. 5b) as reported by others[12,13,58]. We also observed an induction of p16 expression (Supplementary Fig. 5c), indicating that cells have undergone oncogene-induced senescence (OIS).

**Fig. 4** Reduction of Claspin and Timeless levels in HCT116 cells increases spontaneous replication stress and slows down fork progression.
**a** Immunofluorescence analysis of spontaneous γ-H2AX foci in EdU-positive sh-Ctrl, sh-CLSPN and sh-TIM HCT116 cells. Cells were pulse labeled for 10 min with EdU prior to analysis. Representative images are shown. Scale bar, 5 μm. **b** Quantification of γ-H2AX signal intensity in EdU-positive sh-Ctrl, sh-CLSPN, and sh-TIM cells. **c** Western blot analysis of H2AX phosphorylation on Ser139 in untreated sh-Ctrl, sh-CLSPN, and sh-TIM HCT116 cells. **d** DNA combing analysis of replication fork progression in sh-Ctrl, sh-CLSPN, and sh-TIM HCT116 cells. Asynchronous cultures were pulse labeled for 10 min with IdU and 20 min with CldU. DNA fibers were extracted, stretched on silanized coverslips and analyzed by immunofluorescence using antibodies against IdU, CldU, and ssDNA. The length of CldU tracks was determined for three independent experiments. Median lengths are indicated in red. ****$p < 0.0001$, ***$p < 0.001$. Mann–Whitney rank sum test. **e** Exponentially growing sh-Ctrl, sh-CLSPN, and sh-TIM HCT116 cells were labeled for 10 min with IdU, 20 min with CldU and were sorted by FACS according to their DNA content. $G_1$, early S phase (ES), and late S phase (LS) cells were collected for further analysis. **f** Western blot analysis of γ-H2AX levels in sh-Ctrl, sh-CLSPN, and sh-TIM HCT116 cells collected in $G_1$, early S phase (ES), and late S phase (LS). Relative levels calculated for three independent experiments after normalization to tubulin are shown. **g** DNA combing analysis of replication fork speed in sh-Ctrl, sh-CLSPN, and sh-TIM HCT116 cells collected in early S and late S. Median lengths of CldU tracks were determined for three independent experiments and indicated in red. ****$p < 0.0001$, ***$p < 0.001$, ns nonsignificant. Mann–Whitney rank sum test

After 60 days of Ras$^{V12}$ expression, a population of cells escaping senescence was detected which is consistent with earlier studies[59,60]. These cells showed no detectable levels of p16 (Supplementary Fig. 5c), but were unable to form colonies, suggesting that they were not tumorigenic.

Next, we isolated thirteen individual clones from the population of OIS-resistant BJ-Ras$^{v12}$ cells and the expression of ATR, Claspin, Timeless, and CHK1 was analyzed by qRT-PCR. Remarkably, the majority of these clones showed at least a 50% increase in the levels of Claspin and CHK1 mRNAs compared to BJ-hTERT, whereas ATR levels remained unchanged (Fig. 5a). Among them, the clones #4 and #5 overexpressing both Claspin and Timeless were selected for further analyses, together with clone #8, in which these factors were not overexpressed. Western blot analyses confirmed that clones #4 and #5 (but not clone #8) contained increased protein levels of Claspin, Timeless, and CHK1, whereas levels of ATR and Rad17 did not vary between the three clones (Fig. 5b). Interestingly, all three clones displayed increased levels of phospho-CHK1 and γ-H2AX compared to BJ-hTERT cells, indicating the persistence of an oncogene-induced RS (Fig. 5c). Yet, these clones grew more rapidly than BJ-hTERT or BJ-Ras$^{V12}$ cells, suggesting that they have acquired the capacity to tolerate chronic RS (Fig. 5d).

Since the primary manifestation of RS is a slowdown of replication forks, we next monitored fork progression in clones #4, #5, and #8 by DNA fiber spreading. Remarkably, the slow fork progression induced by Ras$^{V12}$ was completely suppressed in clone #4 and fork progression was even faster in clone #5 than in unchallenged BJ-hTERT cells (Fig. 5e). These data suggested that clones #4 and #5 have acquired new properties allowing them to replicate efficiently in presence of RS. Interestingly, clone #8 showed an impaired fork progression (Fig. 5e), suggesting that these cells have adopted a different strategy than clones #4 and #5 to tolerate RS. To identify the pathway(s) potentially involved in this adaptation process, we analyzed the transcriptome of clones #4, #5, and #8 and compared it to BJ-hTERT and BJ-Ras$^{V12}$ cells. Extensive analysis of differentially expressed genes with GO-TERM, KEGG, and REACTOME failed to identify obvious patterns that could explain their phenotypes. Since resistance to RS depends on E2F transcription[61], we have performed a differential expression analysis was performed using a SAM multiclass analysis focusing on different set of genes induced by E2F or E2F hyperactivation (siE2F6), HU exposure[62], and genes involved in DNA repair (http://repairtoire.genesilico.pl). Among these, only DNA repair genes allowed a good separation between clone #4 and #5 and BJ or BJ-Ras cells, clone #8 showing intermediate profiles (Supplementary Fig. 5d). These data suggest that other overexpressed factors contribute to RS tolerance, besides Claspin and Timeless.

Since clone #4 and #5 overexpress a large number of replication factors besides Claspin and Timeless (Fig. 5f, Fold change > 1.5; FDR = 0), we next asked whether depletion of only Claspin or Timeless would be sufficient to prevent RS tolerance, as shown above for HCT116 cells (Fig. 4d). To address this possibility, clone #4 and #5 were transfected with siRNA pools against Claspin and Timeless (Fig. 6a) and fork progression was monitored by DNA fiber spreading. This analysis confirmed the slow fork phenotype induced by the expression of Ras$^{V12}$ in BJ cells and the ability of clones #4 and #5 (but not #8) to restore a normal fork progression despite the induction of Ras$^{V12}$ (Fig. 6b). Remarkably, this adaptation was lost upon depletion of Claspin or Timeless in clones #4 and #5, as manifested by slower fork progression (Fig. 6b) and increased γ-H2AX levels (Supplementary Fig. 6a). These data suggest that the overexpression of Claspin and Timeless is a spontaneous event that is selected at early stages of the cancer process to protect cells from chronic RS.

To determine whether the overexpression of Claspin or Timeless is not only necessary but also sufficient to promote resistance to oncogene-induced RS, these proteins were expressed in BJ-Ras$^{v12}$ (Supplementary Fig. 6b) and in U2OS-CycE cells (Fig. 6c) and their ability to restore normal fork progression was monitored by DNA fiber spreading after induction of Ras$^{V12}$ in BJ-hTERT cells or CycE in U2OS cells. In BJ-Ras$^{v12}$ cells, the overexpression of Claspin, Timeless, or both proteins rescued fork slowdown induced by Ras$^{V12}$ in BJ-hTERT cells (Supplementary Fig. 6c) and by CycE in U2OS cells (Fig. 6d). Together, these data indicate that increased levels of Claspin or Timeless is sufficient to protect cells from oncogene-induced RS.

## Discussion

Oncogene-induced RS plays an active role in tumorigenesis by promoting genomic instability and by inducing a selective pressure for the inactivation of DDR factors such as TP53 and ATM[1]. Spontaneous DNA damage and genomic instability are caused by DNA replication problems occurring in pretumoral lesions[57]. However, DDR signals decrease during cancer development, as illustrated in bladder cancer[2]. Recent evidence indicates that this decreased DDR signaling correlates with increased expression of the checkpoint mediator Timeless in bladder tumors[25]. However, the mechanism by which Timeless promotes adaptation to oncogene-induced RS is currently unclear.

Here, we have monitored the expression of Claspin and Timeless in primary breast, colorectal, and lung cancers and compared it to other components of the ATR-CHK1 pathway. This analysis revealed that Claspin, Timeless, and CHK1 are overexpressed in a coordinated manner in these three cancers whereas the checkpoint sensors ATR, RAD17, and RAD9 are not. We confirmed that Claspin and Timeless protein levels are much

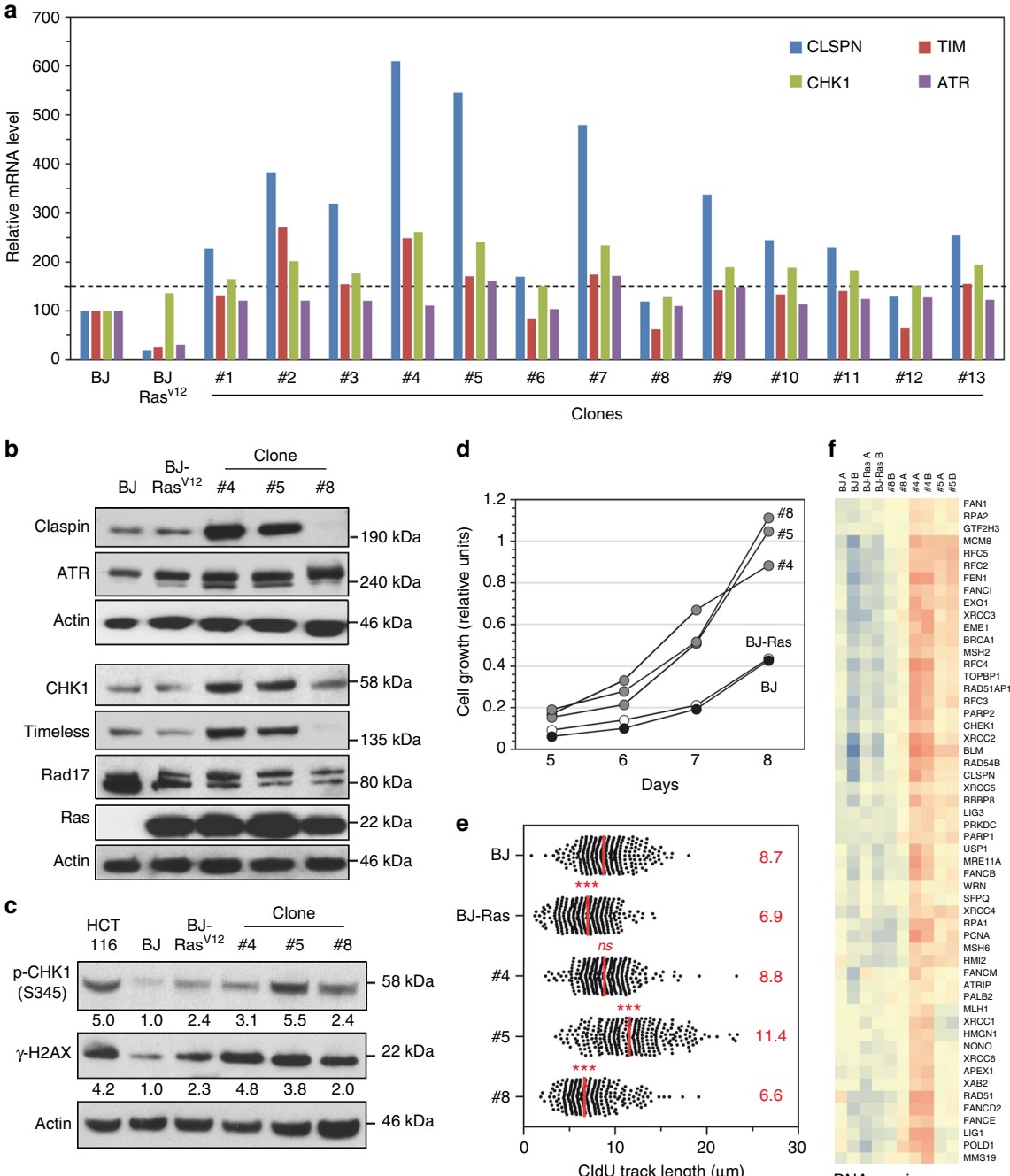

**Fig. 5** Primary fibroblasts adapt to oncogene-induced replication stress by overexpressing Claspin and Timeless. **a** Immortalized BJ-hTERT cells expressing an oncogenic version of Ras (Ras[v12]) under the control of a doxycyclin-inducible promoter were grown for 60 days in the presence of Dox to induce oncogene-induced senescence (OIS). Thirteen individual clones escaping senescence were isolated. The expression of Claspin, Timeless, CHK1, and ATR mRNAs was determined by qRT-PCR in these clones and in BJ-hTERT cells expressing or not Ras[V12] after normalization of mRNA levels to HPRT and 18S. BJ-hTERT and BJ-Ras[V12] cells were grown in presence of doxycyclin 8 days before analysis. **b** Western blot analysis of Claspin, Timeless, CHK1, ATR, RAD17 protein levels in BJ-hTERT, BJ-Ras[v12] cells, and in clones #4, #5, and #8. **c** Western blot analysis of phospho-CHK1 (S345) and γ-H2AX in BJ-hTERT and BJ-Ras[v12] cells and in clones #4, #5, and #8. **d** DNA fiber spreading analysis of fork progression in these cells after 10 min IdU and 20 min CldU pulses. The median length of CldU tracks (red) was determined for five independent experiments. ****$p < 0.0001$, ***$p < 0.001$, ns nonsignificant difference with BJ cells. Mann–Whitney rank sum test. **e** Growth of BJ-hTERT (black), BJ-Ras[v12] (white) after 8 days in the presence of doxycyclin and clones #4, #5, and #8 (gray). **f** Hierarchical clustering analysis of BJ-hTERT, BJ-Ras[v12], and clones #4, #5, and #8 based on the expression of DNA repair genes (in two independent experiments: A and B). Blue: low expression, red: high expression

higher in cancer cell lines relative to immortalized fibroblasts, which is consistent with earlier studies[63,64]. Moreover, we observed a tight correlation between the abundance of Claspin, Timeless, and CHK1 proteins the proteomic landscape of 50 colon cancer cell lines[43]. Together, these data indicate that the

downstream components of the ATR-CHK1 pathway define a functional module that is specifically enhanced in cancer cells.

Importantly, we also found that in low-grade NSCLC, increased expression of Claspin, Timeless, and CHK1 (but not ATR, RAD9, and RAD17), correlates with the aggressiveness of

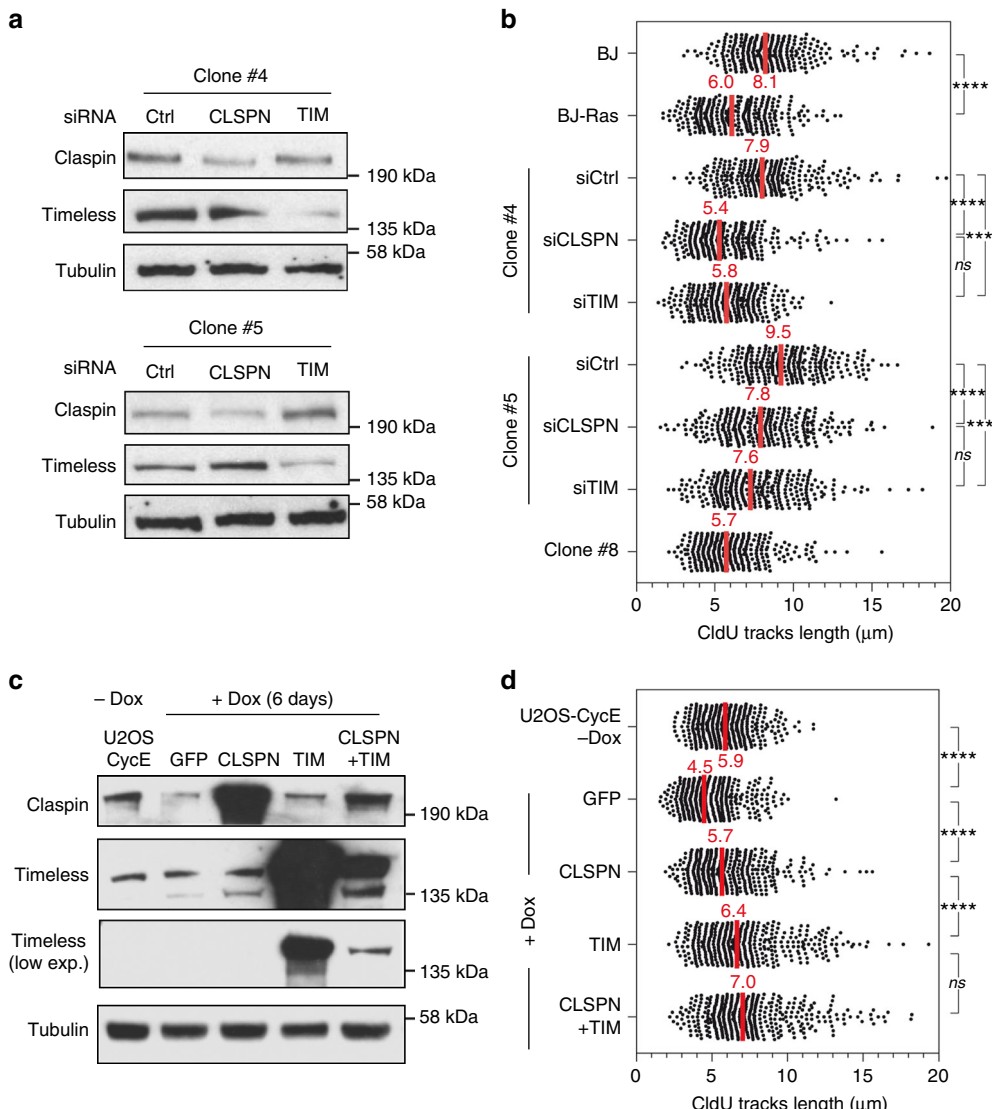

**Fig. 6** Increased levels of Claspin and Timeless promote fork progression in cells expressing oncogenic Ras. **a** Protein levels of Claspin and Timeless in clones #4 and #5 transfected with control (si-Ctrl), Claspin (si-CLSPN), and Timeless (si-TIM) siRNAs. **b** DNA fiber spreading analysis of fork progression in clones #4 and #5 after depletion of Claspin and Timeless with siRNAs. Median CldU tracks length (red) were determined for three independent experiments. ****$p < 0.0001$, ***$p < 0.001$, ns nonsignificant difference. Mann–Whitney rank sum test. **c** Levels of Claspin and Timeless in U2OS-CycE cells transfected with GFP, Claspin, and Timeless plasmids. U2OS-CycE cells expressing Cyclin E under the control of a doxycyclin-inducible promoter were grown during 6 days in the presence of Dox to induce replication stress. Then cells are transfected with plasmids to overexpress GFP, Claspin or Timeless during 3 days before performing DNA spreading experiment. **d** DNA spreading analysis of experiment performed in **c**. DNA spreading was also performed on U2OS-CycE cells growing without Dox. Median CldU tracks length (red) were determined for three independent experiments. ****$p < 0.0001$, ns nonsignificant. Mann–Whitney rank sum test

the tumor. This finding is important because surgical resection is the standard of care for NSCLC and recurrence after resection has been reported in 30–75% of all cases from stage I to stage III[65]. The current clinical staging based on anatomic or pathological factors is insufficient to predict the evolution of the disease, especially for early stage diseases. Our data indicate that Claspin and Timeless could be used as prognostic markers to identify patients who would benefit from adjuvant chemotherapy.

Claspin and Timeless have a dual role in checkpoint signaling and in the maintenance of replication fork integrity[34–36]. To determine which of these two activities promotes RS tolerance, we have reduced the expression of Claspin and Timeless to levels that impede cell growth without interfering with ATR-CHK1 signaling. Conflicting results have been published regarding the impact of Claspin and Timeless depletion on CHK1 activation[10,66–69].

Our data indicate that residual levels of Claspin and Timeless are sufficient to activate CHK1. Alternatively, other checkpoint mediators such as TopBP1 and BRCA1 could be redundant with Claspin and Timeless[33,70]. Our data indicate that the abundance of TopBP1 and BRCA1 correlates with Claspin, Timeless, and CHK1 in colon cancer cell lines, which would be consistent with the latter hypothesis. In any case, we report here that the reduction of Claspin and Timeless levels in cancer cell lines reduced the speed of fork progression, increased the rate of fork stalling and increased γ-H2AX levels in checkpoint-proficient cells. We, therefore, propose that enhanced levels of Claspin and Timeless protect cancer cells from endogenous RS in a checkpoint-independent manner.

The mechanism by which an excess of Claspin and Timeless promote RS tolerance is currently unclear. A large body of

evidence indicates that these two proteins form a FPC together with Tipin and interact with multiple components of the replisome to maintain the integrity of stressed forks[32,34]. This structural function of the FPC could help forks progress through regions of the genome that are intrinsically difficult to replicate, especially in the context of deregulated oncogenic pathways. Interestingly, we have found that Timeless depletion affected preferentially late-replicating regions of the genome, whereas Claspin depletion impacted replication throughout S phase. These data suggest that Claspin and Timeless have nonredundant functions in the FPC, which is consistent with yeast data. Indeed, the budding yeast orthologue of Timeless is required for stable pausing at specific replication barriers, whereas the orthologue of Claspin is not[39,51]. Moreover, Timeless facilitates fork progression through telomeric and pericentric chromatin[71,72] and stimulates the activity of the DDX1 helicase on structured DNA substrates that could be more abundant in late-replicating, repeated regions of the genome[73]. Interestingly, recent evidence indicates that Timeless is displaced from the replisome under RS conditions to slowdown fork progression and prevent deleterious consequences for the integrity of the genome[74]. In this context, enhanced levels of Claspin, Timeless, and Tipin in cancer cells may promote the reassembly of the FPC and accelerate fork restart. This is consistent with the fact that overexpressed Claspin and Timeless accumulate on chromatin in HCT116 cells and could be easily available to reform the FPC upon stress-induced dissociation. Interestingly, the overexpression of Timeless in bladder cancer is associated with increased genomic instability[25]. It is, therefore, likely that the adaptation to oncogene-induced RS mediated by enhanced Claspin and Timeless occurs at the expense of genome integrity and would therefore promote cancer progression.

Another important question raised by our study concerns the mechanism by which cancer cells overexpress Claspin and Timeless. The expression of Claspin and CHK1 depends on the E2F pathway[61,75], which are upregulated by oncogenes and contributes to RS tolerance[61]. To determine whether the upregulation of the E2F pathway is the initial event contributing to the upregulation of Claspin and Timeless, we have overexpressed an oncogenic form of Ras in immortalized primary fibroblasts and monitored the expression of Claspin, Timeless in clones escaping OIS[12,13]. Remarkably, a large fraction of these clones showed increased levels of Claspin and to a lesser extent Timeless. We also detected increased levels of CHK1 mRNAs, but not ATR, which is consistent with the levels observed in tumor samples and cancer cell lines. As in cancer cell lines, the depletion of Claspin and Timeless in these clones increased oncogene-induced RS, stressing the importance of these proteins the adaptation to stress. Transcriptome analyses of these clones revealed complex and heterogeneous gene-expression profiles that are not consistent with a simple upregulation of the E2F pathway, but with a global overexpression of a large group of DDR genes involved in fork protection. Interestingly, one of the OIS-resistant clones did not overexpress Claspin and Timeless, indicating that alternative pathways exists to adapt to RS.

In conclusion, our work supports a model in which the overexpression of Claspin and Timeless protects forks from chronic RS by stabilizing the replisome without increasing the activation of the ATR pathway. This would confer a proliferative advantage to cancer cells experiencing chronic RS by promoting replication fork progression without triggering an increased checkpoint response, which would be detrimental to tumor growth. These data indicate that Claspin and Timeless represent promising targets for anticancer therapies targeting replication forks. Moreover, they represent valuable prognostic markers for patients with early NSCLC.

## Methods

**Tumor samples, RNA extraction, and statistical analysis**. RNA samples were extracted and quantified from 93 untreated primary lung adenocarcinomas, 206 primary breast cancers (PACS01 trial), and 74 primary colorectal carcinomas as described[27,41,42]. Written informed consent was obtained from all patients before testing and experiments conformed to the principles set out in the WMA Declaration of Helsinki. DFS data were available for 72 low-stage (IA/IB/IIA/IIB) lung cancer patients. All survival times were calculated from the date of surgery and were estimated by the Kaplan–Meier method with 95% confidence intervals (CI), using the following first-event definitions: loco-regional relapse, distant metastasis, other cancer or death from any cause. Patients alive without disease are censored at the date of last follow-up. Gene-expression values were divided into two equal groups by taking the median value. Univariate analysis was performed using the Log-rank test. Two-sided $p$ values of less than 0.05 were considered statistically significant. Statistical analyses were performed using STATA 12.0 software.

**Immunohistochemistry**. Paraffin-embedded specimens were obtained from 14 breast cancer patients for immunohistochemical analysis. The procedures of immunostaining were carried out with standard protocols by the PETRA-AMMICa Facility (Gustave Roussy Institute, France). Briefly, 3 μm paraffin tissue slides were deparaffinized after antigen retrieval and quenching of endogenous biotin, sections were incubated at room temperature for 60 min with a rabbit anti-Timeless antibody (1/100, ab109512, Abcam). Antibody binding was detected using the EnVision Flex Dako kit (Agilent) and counterstaining was performed with Mayer's Hemalun solution.

**Cell lines, cell culture, and drugs**. The human HCT116 colorectal carcinoma cell line was provided by A. Coquelle (IRCM, Montpellier, France). Normal human fibroblasts IMR90 and BJ-hTERT were a gift of J-M Lemaitre (IGF, Montpellier, France). MCF7 breast cancer cells were provided M. Piechaczyk (IGMM, Montpellier, France). HeLa cervical cancer cells and U2OS cells were a gift of M. Benkirane (IGH, France), All cell lines were grown in high glucose Dulbecco's modified Eagle's medium with ultraglutamine, supplemented with 10% heat inactivated fetal bovine serum (FBS) and 10% antibiotics (Lonza). Cell lines were grown at 37 °C in a humidified atmosphere of 5% $CO_2$ and were tested for absence of mycoplasma contamination.

**Correlation analysis of proteomics data**. Colorectal cancer cell line proteomics data were retrieved from ref. [43]. Correlation heatmaps were generated with the Morpheus clustering web tool (https://software.broadinstitute.org/morpheus/) using Pearson's correlation.

**Cell line transduction and plasmid transfection**. pBABE-Puro H-RasV12 (n° 12545 Addgene) or Cyclin E and corresponding empty vectors were used to infect either BJ hTERT cells or U2OS, respectively. Lentiviruses were produced by the vectorology facility (BioCampus Montpellier) according to standard protocols[76]. Plasmid transfection in BJ-Ras[v12] cells was performed using Viromer Yellow (Lipocalix) according to the manufacturer's instructions. U2OS-CycE transfections were performed using Interferin (Polyplus-transfection, Illkirch, France) according to the manufacturer's instructions. As a control, pMax-GFP plasmid (Lonza) was used and to overexpressed Claspin and Timeless pcDNA3.1-Flag-claspin (Addgene no. 12659) and pcDNA 4-Flag-Timeless (Addgene no. 22887) plasmids were used.

**RNA interference and transfection**. MISSION® TRC shRNA plasmid targeting Claspin (TRCN0000130853) and Timeless (TRCN0000157801) and MISSION® pLKO.1-puro Non-Mammalian shRNA Control Plasmid (shc002) (Sigma-Aldrich) were used to produce lentiviruses expressing constitutive shRNA. TRIPZ inducible lentiviral shRNA plasmid (Thermo Scientific Open Biosystems) targeting Claspin (V2THS_200492) and Timeless (V2THS_47526) were used to produce lentiviruses expressing inducible shRNAs on the vectorology facility (BioCampus Montpellier)[76]. HCT116 cells were transduced with these different lentiviruses expressing individual shRNA and heterogeneously-infected population of cells expressing shRNA was selected with 1 μg/ml Puromycin. Inducible shRNA are expressed by addition of 2 μg/ml Doxycycline for three days before experiments. Infected cell populations were maintained in culture for no more than one month. For siRNA transfection, OnTarget plus SMART pool (Dharmacon) against Claspin (L-005288-00) or against Timeless (L-019488-00) were used. Hiperfect reagent (Qiagen) has been used for HCT116 cells transfection and Amaxa cell line Nucleofector Kit R technology (Lonza) for clones 4 and 5 transfection.

**Immunoblotting**. Cells were lysed in 2× Laemmli buffer at the concentration of $1 \times 10^4$ cells/μl. Lysates were treated with 3 μl of Benzonase (25 U/μl, Sigma) for 30 min at 37 °C. RPE-1 and MCF10A cells extracts were a gift of M. Mechali and R. Fernandez de Luco labs respectively (IGH, France). Proteins were separated by sodium dodecyl sulfate polyacrylamide gel electrophoresis, transferred to nitrocellulose membrane and analyzed by western immunoblotting with appropriate antibodies: anti-Claspin (1/50, gift of T. Halazonetis, Geneva), anti-Timeless

(1/1000, Interchim), anti-Actin (1/500, Sigma), anti-phospho Ser317-CHK1 or Ser345-CHK1 (1/1000, Cell Signaling Technology, ref 2344S and 2348), anti-CHK1 (1/1000, Cell Signaling Technology, ref 2360), anti-Cdc25A (1/200, Santa Cruz, ref sc-7389), anti-γ-H2AX (1/1000, Millipore, ref 05–636), anti-Tubulin (1/3000, Abcam, ref ab-6161), anti-RPA (1/300, Abcam ref ab-79398), anti-ATR (1/1000, Abcam, ref ab-10312), anti-RAD17 (1/1000, MBL, ref K0120-03), anti H3 (1/3000, Abcam, ref ab-7191), anti-ras (1/500, BD, ref 610002), anti-Actin (1/500, Sigma ref A4700). Blots were incubated with horseradish peroxidase-linked secondary antibody (GE Healthcare) and visualized using the ECL$^+$ chemiluminescence method (Pierce).

**Cell fractionation**. Cells were lysed with 0.2% Triton X-100 CSK buffer (10 mM PIPES [pH 6.8], 100 mM NaCl, 300 mM sucrose, MgCl$_2$, 1 mM EGTA, 1 mM EDTA, proteases inhibitors cocktail (Complete, EDTA-free tablets; Roche), and 0.2% Triton X-100) for 10 min on ice. After centrifugation ($0.8 \times g$ for 5 min), the supernatant is the soluble fraction (sup). The pellet was washed once more with the same buffer and incubated 10 min on ice. After centrifugation ($0.8 \times g$ for 5 min) the pellet (chromatin fraction) was resuspended in 2× Laemmli buffer.

**Cell proliferation quantification**. The Cell Proliferation Reagent WST-1 (Sigma) is used for the spectrophotometric quantification of cell proliferation.

**Soft agar assay**. Totally, $5 \times 10^4$ cells were resuspended in 3 ml of prewarmed DMEM-10% FBS and were mixed with 3 ml of 0.8% agar solution (Sigma A5431) in 10% DMEM and 1% FBS. A total of 1.5 ml of cell suspension is poured in three wells containing a 2 ml prewarmed layer of 1% Agar in 10% DMEM and 1% FBS. Totally, 1 ml of medium is added on the upper agarose layer and plates are incubated for 15 days at 37 °C. Top medium is replaced every three days. Colonies were visualized with a microscope and colony sizes were determined with the MetaMorph software.

**Colony formation assay**. HCT116 cells are plated in 6-well plates at 20% confluence and are grown for 2 weeks. Cells are washed twice with ice-cold phosphate-buffered saline (PBS) and fixed for 10 min with ice-cold methanol. Colonies are stained for 10 min at room temperature with 0.5% crystal violet in 25% methanol and rinsed with ddH20.

**Cell sorting**. After trypsinization, asynchronous cells were resuspended in PBS-3mM EDTA ($10^7$ cells/500 μL cells) and fixed in ETOH 70% as in the previous section. After two washes in ice-cold PBS, cells were resuspended at the concentration of $5 \times 10^6$ cells/ml in staining buffer (3 mM EDTA, 0.05% NP40, 50 μg/ml propidium iodide, 1 mg/ml RNase A in PBS). Cells in G$_1$, early S, and late S phase were collected according to their DNA content with a FACS-Aria cell sorter (Becton Dickinson) at the IGMM-MRI facility (Montpellier, France). Immediately after collection, each fraction of cells was divided in two parts: one for proteins analysis by western blot and one for DNA combing.

**Flow cytometry analysis of γ-H2AX and EdU**. Asynchronous cells were pulse labeled with 10 μM EdU (5-ethynyl-2′-deoxyuridine) for 30 min before harvesting. Cells were trypsinized as usual, washed once with cold PBS and once with PBS–bovine serum albumin (BSA) 1%. A total of $1 \times 10^6$ cells were fixed in 2% formaldehyde–PBS for 10 min in the dark at room temperature and washed twice in 1 ml of cold PBS/1% BSA. cells were permeabilized by addition of 200 μL of saponin buffer (0.1% saponin in PBS/0.5% BSA) for 10 min at 4 °C and incubated with γ-H2AX antibody (Millipore 05-636) diluted at 1/500 in 100 μL of saponin buffer for 2 h at room temperature in the dark. After two washes in saponin buffer cells were incubated in 50 μL of saponin buffer containing the secondary anti-mouse Alexa 488 antibody (Molecular probes: A-11001) at 1/500 for 30 min at room temperature in the dark. Cell were washed twice in 0.5 ml of saponin buffer and resuspended in 100 μL of Click-it reaction mix (86.5 μl H$_2$O, 3 μl CuSO$_4$ (0.1 M), 0.5 μl Alexa fluor 647-azide (Thermo Fisher Scientific), 10 μl of 100 mM freshly prepared vitamin C) for 30 min at room temperature in the dark. Cells were washed twice with 0.5 ml PBS/1% BSA and resuspended in 0.5 ml 1% BSA/PBS containing 0.1 mg/ml RNase A and 2 μg/ml DAPI. After 30 min at 37 °C samples were analyzed with the Miltenyi Macs Quant analyzer and FlowJo software at the IGH-MRI facility (Montpellier, France).

**Single-nucleotide polymorphism comparative genomic hybridization**. CNVs in sh-Ctrl, sh-CLSP, and sh-TIM cells were determined using Illumina Infinium technology, using BeadChips (HumanHap300 duo) containing 317000 SNP tags. The cnvPartition plug-in of the Illumina BeadStudio software was used to annotate CNVs and estimate copy number.

**DNA combing and DNA fiber spreading**. DNA combing and DNA fiber spreading were performed as described[48,49,77] using the following combination of antibodies. *Primary antibody mix:* Mouse anti-BrdU to detect IdU (1/100 PBS/T, BD 347580), Rat anti-BrdU to detect CldU (1/100 PBS/T, Eurobio clone BU1/75),

Anti-ssDNA (1/100 PBS/T, auto anti-ssDNA, DSHB). *Secondary antibody mix:* Goat anti-rat Alexa 488 (1/100 PBS/T, Molecular Probes, A11006); Goat anti-mouse IgG1Alexa 546 (1/100 PBS/T, Molecular Probes, A21123); Goat anti-Mouse IgG2a Alexa 647 (1/50 PBS/T, Molecular Probes, A21241). Statistical analysis of track lengths is performed with GraphPad Prism 7.0.

**Immunofluorescence**. HCT116 cells were seeded on cover glasses and labeled with 10 mM EdU for 15 min. Cells were then fixed with 2% paraformaldehyde and permeabilized using 0.1% Triton X-100. Immunodetection of γH2AX in S-phase was performed overnight at 4 °C using a specific antibody (1/100; 05-636, Millipore). Click-it chemistry was performed according to instructions. Images were acquired using an LSM780 Zeiss confocal microscope. Mean fluorescence intensity of γ-H2AX in EdU-positive cells was quantified with CellProfiler.

**Transcriptome analysis**. RNA sequencing (RNA-seq) libraries were prepared using the Illumina TruSeq Stranded mRNA Library Prep Kit. Paired-end RNA-seq were performed with an Illumina NextSeq sequencing instrument (Helixio, France). RNA-seq read pairs were mapped to the reference human GRCh37 genome using the STAR aligner[78]. Statistical analyses were performed with R (v3.2.3) and R packages developed by BioConductor project. The expression level of each gene was summarized and normalized using DESeq2 R/Bioconductor package[79]. Differential expression analysis was performed using SAM multiclass analysis (samr package). Genes with a fold change > 1.5 and a FDR < 0.05 were considered as significantly differentially expressed.

**Reporting summary**. Further information on experimental design is available in the Nature Research Reporting Summary linked to this article.

## Data availability
The datasets generated during and/or analyzed during the current study are available from the corresponding author on reasonable request. The RNA-seq datasets generated and analyzed during the current study (Fig. 5f and Supplementary Fig. 5d) are available in the GEO repository, accession number: GSE123380. A reporting summary for this Article is available as a Supplementary Information file.

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

## Acknowledgments

We thank Vasilis Gorgoulis (University of Athens) and Thanos Halazonetis (University of Geneva) for providing the anti-Claspin monoclonal antibody. We thanks Benjamin Pardo for discussions and critical comments on the manuscript. We thank E. Schwob and the DNA combing facility of Montpellier for providing silanized coverslips. We thank the Montpellier RIO Imaging microscopy and cell sorting facility for help with image analysis. We thank J.-J. Maoret (GeT-TQ facility, Genopole Toulouse Midi-Pyrénées) for help with qRT-PCR analyses. We thank A. Monteil and C. Lemmers from the Vectorology facility, PVM Biocampus Montpellier, CNRS UMS3426. Work in the Pasero lab is funded by ANR, INCa, MSDAvenir Fund, SIRIC Montpellier Cancer (grant INCa Inserm DGOS 12553) and by Ligue contre le Cancer (équipe labellisée). Work in J.S.H. laboratory is supported by funding from INCa-PLBIO 2016, ANR PRC 2016, Labex Toucan and La Ligue contre le Cancer (Equipe labellisée). JB thanks the Fondation ARC and the French Ministère de la Recherche et de l'Enseignement Supérieur for support.

## Author contributions

J.N.B., V.B., M.-J.P., Y.-L.L., A.-L.S. and H.T. performed the experiments. J.G., A.L. and J.M. analyzed gene expression and clinical data. M.L.T. performed immunochemistry experiments. J.M. analyzed transcriptomic data. T.I.R. and J.C. perform proteomic analysis. J.-S.H., H.T. and P.P. designed the experiments. H.T. and P.P. wrote the manuscript.

## Additional information

**Competing interests:** The authors declare no competing interests.

