## [Peer Review File · Nature Communications]

Reviewers' comments:

Reviewer #1 (Remarks to the Author):

This dataset by Bianco et al. addresses a relevant and important question, and presents some observations and results of potentially broad interest. On the other hand, there is a great deal of confirmatory data or data offering only modest advance and little mechanistic insight. There are also some controls missing in the manuscript, making interpretation of the results more difficult. Overall, the manuscript is mainly descriptive and the novelty and level of experimental rigor are presently not strong enough for a publication in high-impact journals.

Specific points

1. Limited advance of the field: Several of the major points presented here have been published, some by the authors themselves such as the facts that claspin and timeless are overexpressed in some tumors or tumor cell lines, and in clinical samples, this correlates with adverse prognosis- even the lung cancer section shown here as a culmination of the results is mostly just confirmatory with the data published by the authors themselves before: by Allera-Moreau et al. *Oncogenesis*, 2012). The interesting new observations are then left at a descriptive level.

Other authors already reported the same concept of e.g. Tim/Tipin needed for the maintenance of DNA replication fork movement in the absence of damage/independently of the checkpoint: Kaufmann group. *Mol Cell Biol.* 27: 3131-3142, 2007 (not cited here !!); or promotion of normal replication fork rates in human cells by claspin (Petermann et al., *Mol Biol of the Cell* 19: 2373-2378, 2008)- Petermann et al even used exactly the same cell line: HCT116, almost ten years earlier to show this result !.

Thus, to sufficiently boost the enthusiasm for publication, i suggest that two aspects of this work can/should be developed mechanistically:

- i) The issue of how do claspin and timeless mRNAs induced upon replication stress - i.e. how does signaling of replication stress translate into activation of which transcription factors that upregulate the transcription of Claspin and Timeless?
- ii) Mechanistic insights into how, at the molecular level, do CLS and/or TIM protect replication forks independently of their checkpoint roles.

Such mechanistic insights are completely lacking here, and are important to really better show how cells adapt to replication stress, an issue that is the focus of this work and highlighted by the title.

2. The main data rely on replication for labeling. However, the interpretation of these results is difficult as the authors only show one of the two pulses, rather than both the first and the second pulse. It would be important to use the standard approach and show measurements of both CldU and IdU DNA fibers, to judge the important parameters of the degree of fork stalling and collapse, for example.

3. Another problem is the lack of appropriate normal human cell types to compare the protein expression levels with those in carcinoma and osteosarcoma cell lines used in the experiments. Normal human proliferating epithelial cells (widely available), as a very minimum, should used besides the fibroblasts.

4. Only mRNA levels of CLSP, TIM etc. are examined in the clinical tumor material. As the protein levels often do not correlate with mRNA levels, it would be important to perform some protein analysis of a cohort of tumors - either by western blots or well-controlled immunohistochemistry, as it is the proteins that carry out the function.

5. The experiments with the HCT116 colon cancer line are useful, however these cells are really exceptional even among tumor cell lines due to their exceptional recombination activity for which they have been used for gene knockouts in the past. It would be reassuring if at least one additional cancer cell line was used in parallel, to validate the main results and show they are representative.

6. Figs 2A,B, and S2A lack any comparison with normal cells.

7. It is concluded that the chromatin (fork)-binding activity of Claspin and Timeless are increased under replication stress, however this conclusion is not sufficiently documented. What is needed is the ratios between the levels of these proteins in supernatants and chromatin fractions of normal and cancer cells with and without hydroxyurea treatment (e.g. Fig. S1C). It looks as if the ratios are not appreciably different, only the overall levels, which is a result known for years. I.e., an alternative interpretation could be that the chromatin binding potential is NOT enhanced but is the same as in normal cells and merely parallels the overall level of the protein.

8. It was unclear what was the level of phospho-Chk1 in Fig. 4C, as the panel was entirely invisible, at least in the copy I got for review.

9. There are data on the adapted clones (p. 11, end of the penultimate paragraph) quoted as not shown. This is not acceptable and such data should be shown, albeit as a supplement.

10. Fig 4F and the text at the bottom of p. 11 lack inclusion of clone 8 for consistency, and also control blots to assess the extent of siRNA mediated depletion of Tim and CLSP.

11. As mentioned briefly above, the manuscript culminates by a section on NSCLC patients (p. 12), however this result, namely that levels of Tim/CLSP correlate with disease-free interval, was already published by these authors (Allera-Moreau et al. *Oncogenesis*, 2012) and therefore it is merely a confirmatory result.

12. There seems to be a discrepancy between the data in this manuscript versus those in the papers by Petermann in 2008 and the Kaufmann lab in 2007 (see refs above), in that the latter authors both observed decreased phospho-Chk1 under stress in CLSP/TIM-depleted cells, which here it is concluded that there is no change. Can the authors suggest some explanation for such discrepancy, also because the same cell line was used in some of these older studies.

Reviewer #2 (Remarks to the Author):

This study by Bianoc et al. reports that selected primary cancers and cancer cells specifically overexpress the replication stress response factors Claspin, Timeless and Chk1, but not other central components of the replication stress response such as ATR. This suggests that elevated levels of the former proteins are important for the proliferation of cancer cells. The study then concentrates on Claspin and Timeless and shows that these promote the proliferation of cancer cells and this is independent of ATR signalling through Chk1 and Cdc25A. Interestingly, oncogenic Ras-expressing fibroblasts that have escaped senescence also have increased levels of Claspin, Chk1 and Timeless, suggesting that these proteins can allow cells to proliferate in presence of oncogene-induced replication stress. Finally the authors suggest that overexpression of these proteins correlates with poor outcomes in cancer patients and that they could pose potential targets for cancer therapy.

The data are of good quality and provide new insights are potentially of interest to the wider field. Particular strengths are the in-house analysis of tumour using qRT-PCR (rather than just high-throughput data) in Figure 1 and the generation and analysis of Ras-induced senescence resistant

clones in Figure 4.

However it has been reported previously that depletion of Claspin, Chk1 or Timeless reduces replication fork progression in cancer cells, including HCT116 cells (refs 43,44 cited by the authors and Unsal-Kacmaz K, Mol Cell Biol 2007). The independence of Claspin function from Chk1-Cdc25A has also been reported (ref 44). The novelty of this study therefore rests on whether it can make a convincing case that maintenance of replication fork progression by these proteins is more important for proliferation of cancer cells than normal cells. There are some questions in whether this has been achieved. Further evidence will be required to strengthen this conclusion.

Main points:

- Is there evidence that the increased Chk1, Claspin and TIM mRNA levels in primary lung, breast and colorectal cancers translate into increased protein levels (Figure 1)?

- Are Claspin and Timeless only required for the proliferation of HCT116 cancer cells or would the same effects happen in normal cells (Figure 2, Figure 3)? The text argues that HCT116 cells still have more Claspin and Timeless after shRNA depletion than untreated IMR90 cells, yet their growth is still impaired. But has this been normalised for S phase content?

- Figure S1 shows that HCT116, HeLa and U2OS cells have larger S phase population than IMR90 cells. It is mentioned in the text that protein levels have been normalised to S phase fraction, but these data should be shown. Normalisation of other signals such as gamma-H2AX should also be shown.

- If increased fork speeds due to Claspin and TIM overexpression allow cells to adapt to Ras-induced replication stress, then overexpressing these proteins in BJ-Ras cells should protect those from senescence (Figure 4). Can this be tested?

- What is the effect of Claspin and TIM depletion on growth of clones 4, 5 and 8 (Figure 4)? What is the effect on DNA damage signals such as gamma-H2AX? It would be much more helpful to have the analyses of Figure 2 and 3 done in the adapted clones versus BJ and BJ-Ras cells.

- Page 6: "The overexpression of CLSPN and CHK1 correlates with expression cell proliferation marker Ki67 in lung adenocarcinomas, but not in breast and colon cancers (data not shown)." Please show these data. Secondly, if CLSPN and CHK1 overexpression correlate with proliferation in lung cancers, then how informative is the effect of this overexpression on disease-free survival in Figure 5?

Other comments:

Page 9: "These data are consistent with results obtained in other cancer cell lines 43,44." Work in ref 43 was in fact performed in HCT116 cell as well. Secondly, Unsal-Kacmaz K, Mol Cell Biol 2007, previously reported fork slowing induced by Timeless depletion and should be cited.

Figure 4C: The phospho-Chk1 blot is not visible in my version of the PDF

Figure S1C: Loading control for chromatin should be added

Figure S2C: The phospho-Chk1 blot is not visible

BJ-hTert cells should not be referred to as primary cells; they are immortalised non-cancer cells, which is not the same.

Reviewer #3 (Remarks to the Author):

This manuscript proposes that cancer cells adapt to oncogene-induced replication stress by

overexpressing claspin and timeless.

My opinion is that the main premise of this manuscript is correct and important. Indeed, in a number of settings, timeless, tipin and claspin have been shown to be important for the response of cells to DNA replication stress. My main concern is novelty, as similar observations have been reported previously. Of course, this manuscript goes beyond what has been previously published, but by the way the manuscript is written, this is not clear. Previous observations made by others should be cited in the Introduction and not in the Discussion. A full list of relevant papers should be cited. And the questions addressed by this manuscript and the advances made should be clearly stated.

Specific Comments

1. Others have previously reported that timeless and claspin are overexpressed in human cancer. The authors cite a few papers in the Discussion, but from the Introduction, it appears that nobody has looked at this before. This should be fixed. The authors should also cite a full list of relevant papers. I quickly found:

- Yoshida, Sato et al, Cancer Science 2013, showing overexpression of timeless in lung cancer and poor correlation with patient survival (not cited by the authors)
- Fu, Leaderer et al., Mol Carcinog 2011, overexpression of timeless in breast cancer and link to risk (not cited)
- Baldeyron et al., Mol Oncol 2015, tipin overexpression in cancer (not cited).

The authors have, however, cited several relevant papers on this point in the Discussion.

2. In Fig 3A, it is difficult to position the gate to identify the gH2AX-positive cells, because the difference between the gH2AX negative and positive signals is small. How many times was this experiment repeated? Perhaps, gH2AX by FACS is not the best assay for replication stress in this system.

3. Fig 3 C and F, depletion of claspin reduces CldU track length in both early and late S phase cells, but depletion of timeless reduces CldU track length only in early S. What is the explanation for this different behavior? How many times has this experiment been done?

4. Fig. 4A. How many times has this experiment been performed? There are no error bars.

5. Have all the experiments with cell lines been performed in triplicate?

Reviewer #1:

This dataset by Bianco et al. addresses a relevant and important question, and presents some observations and results of potentially broad interest. On the other hand, there is a great deal of confirmatory data or data offering only modest advance and little mechanistic insight. There are also some controls missing in the manuscript, making interpretation of the results more difficult. Overall, the manuscript is mainly descriptive and the novelty and level of experimental rigor are presently not strong enough for a publication in high-impact journals.

We are sorry that this Reviewer did not appreciate the novelty of our work. The point was obviously not to demonstrate that Claspin and Timeless have a dual role in the replication stress response (RS), which was already known, but to determine which of these two functions promotes tolerance to RS in cancer cells. We have largely reorganized the manuscript to clarify this issue. We also apologize for the missing controls, which are now included in the manuscript.

Specific points

1. Limited advance of the field: Several of the major points presented here have been published, some by the authors themselves such as the facts that claspin and timeless are overexpressed in some tumors or tumor cell lines, and in clinical samples, this correlates with adverse prognosis- even the lung cancer section shown here as a culmination of the results is mostly just confirmatory with the data published by the authors themselves before: by Allera-Moreau et al. *Oncogenesis*, 2012). The interesting new observations are then left at a descriptive level.

We agree with this reviewer that the overexpression of Claspin and Timeless in tumor samples and cancer cell lines has already been reported and the corresponding studies are properly cited in our manuscript. However, our work represents the first integrated analysis of the expression of the whole ATR-CHK1 pathway by RT-qPCR in three different cancers (**Fig. 1** and **Extended data Fig 1**). Although our data confirmed that Claspin and Timeless are overexpressed in these cancers, they also show that this occurs in a highly correlated manner, independently of the other components of the pathway. We also provide evidence that this correlation also occurs at the protein level in the proteomic landscape of 50 colon cancer cell lines (**Fig. 2c**). Together, these data suggest that Claspin, Timeless and CHK1 define a functional module that is distinct from the rest of the ATR-CHK1 pathway. This information is novel and represents the starting point of the present study.

Regarding the outcome of NSCLC patients (now **Fig. 1d**), it is worth noting that our analysis differs in many ways from the Allera *et al.* study: (i) it is restricted to early stages (I and II) patients who did not receive adjuvant treatment, (ii) it extends over a much longer period (72 months), (iii) it addresses disease-free survival and not overall survival, (iv) it focuses on the ATR-CHK1 pathway. The conclusions of this study are important because 30% to 50% of stage I and IIA patients die within five years of surgical resection. Many of them did not receive adjuvant treatment because of the inability of conventional TNM stratification to identify patients who would benefit from adjuvant chemotherapy. The identification of novel predictive biomarkers of recurrence (including Claspin and Timeless) is therefore an important issue that was not addressed in previous studies.

Other authors already reported the same concept of e.g. Tim/Tipin needed for the maintenance of DNA replication fork movement in the absence of damage/independently of the checkpoint: Kaufmann group *Mol Cell Biol.* 27: 3131-3142, 2007 (not cited here !!); or promotion of normal replication fork rates in human cells by claspin (Petermann et al., *Mol Biol of the Cell* 19: 2373-2378, 2008)- Petermann et al even used exactly the same cell line: HCT116, almost ten years earlier to show this result !

The purpose of our study was not to show that Claspin and Timeless have a checkpoint-independent function in fork maintenance as this was indeed already known. We apologize if this was not clear in the previous version of our manuscript. Yet, we would like to stress the fact that the two references mentioned by the referee were actually cited in our manuscript, including two articles from the Kaufmann group (2005 and 2007) and the Petermann lab in 2008. With that in mind, the question was still to determine which of the two known functions of Claspin and Timeless (checkpoint signaling or fork protection) was important for tolerance to RS. Using levels of depletion that interfere with fork progression without affecting ATR-CHEK1 signaling, we demonstrate that the fork protection function is important for RS tolerance, independently of CHEK1 activation.

Thus, to sufficiently boost the enthusiasm for publication, I suggest that two aspects of this work can/should be developed mechanistically:

i) The issue of how do claspin and timeless mRNAs induced upon replication stress - i.e. how does signaling of replication stress translate into activation of which transcription factors that upregulate the transcription of Claspin and Timeless?

We are grateful to this Reviewer for these constructive comments. To address the first one, we have performed an extensive analysis of the transcriptome of BJ clones that escaped Ras-induced senescence and overexpress Claspin and Timeless. Claspin and Timeless, like many other DNA replication factors, are targets of E2F transcription factors family (Bertoli et al., Cell Reports 2016). Since the E2F pathway is induced in response to RS (Bertoli et al., Cell Reports 2016), we have checked whether it is upregulated in clones overexpressing Claspin and Timeless. However, SAM multiclass analyses of E2F target genes or genes induced by HU failed to efficiently separate cell populations (**Extended data Fig. 5d**). Yet, we detected a global upregulation of DNA repair genes acting at replication forks in clones surviving Ras induction compared to others (**Fig. 5f**). These cells also displayed a high variability in their gene expression profiles. Together, these data suggest that adaptation to RS in BJ-Ras cells cannot be only assigned to the upregulation of the E2F pathway, but results from a high genomic instability and the selection of clones overexpressing critical fork protection factors such as Claspin and Timeless.

ii) Mechanistic insights into how, at the molecular level, do CLS and/or TIM protect replication forks independently of their checkpoint roles.

Such mechanistic insights are completely lacking here, and are important to really better show how cells adapt to replication stress, an issue that is the focus of this work and highlighted by the title.

As indicated above, a large number of studies in yeast and vertebrates (including ours) indicate that Claspin and Timeless interact with multiple components of the replisome and stimulate/coordinate the activity of DNA polymerases and helicases. The aim of our study was not to further explore this role, but to assess its relevance in the context of tolerance to oncogene-induced replication stress. It has been recently reported by the Lukas lab that Timeless disassembles from the replisome in response to replication stress, which slows down replication to prevent fork collapse (Somyajit *et al.*, Science 2017). Here, we show that a large excess of Claspin and Timeless accumulates on chromatin in HCT116 cells (**Extended data Fig. 2**). We propose that this excess promotes a rapid replacement of these components on the replisome to promote fork restart. This is based on the following observations:

- Reducing the levels of Claspin and Timeless in HCT116 under conditions that do not prevent CHEK1 activation decreases fork progression (**Fig. 4d-g**) and increases sister fork asymmetry, which is indicative of increased fork stalling (**Extended data Fig. 4d**).
- Clones overexpressing Claspin and Timeless (clones #4 and #5, but not clone #8) show a normal fork progression in the presence of Ras-induced RS (**Fig. 5e**). Reduction of Claspin or

- Timeless excess in these clones with siRNAs unmasks Ras-induced RS and reduces fork speed to the same level as BJ-Ras cells (**Fig. 6b**).
- The overexpression of Claspin and Timeless in BJ cells overexpressing Ras^{V12} (**Extended data Fig. 6c**) or in U2OS cells overexpressing Cyclin E (**Fig. 6d**) increases tolerance to oncogene-induced RS.

2. The main data rely on replication for labeling. However, the interpretation of these results is difficult as the authors only show one of the two pulses, rather than both the first and the second pulse. It would be important to use the standard approach and show measurements of both CldU and IdU DNA fibers, to judge the important parameters of the degree of fork stalling and collapse, for example.

DNA fiber experiments were performed using standard procedures that we have contributed to establish in numerous publications (Tuduri 2009 Nat Cell Biol, Bianco 2012 Methods, Tourriere 2017 BioProtocol, Coquel 2018 Nature...). To determine fork speed, only the CldU track is measured, the IdU pulse being only used to determine fork polarity and to ensure that the signal correspond to a single ongoing fork. The comparison of IdU and CldU tracks is not a good indication of fork stalling. For this purpose, the standard procedure is to compare the length of CldU tracks at diverging sister replication forks, as illustrated in **Extended data Fig. 4d**. For more details on standard procedures, please refer to Techer *et al.* (2013) J Mol Biol 425, 4845.

3. Another problem is the lack of appropriate normal human cell types to compare the protein expression levels with those in carcinoma and osteosarcoma cell lines used in the experiments. Normal human proliferating epithelial cells (widely available), as a very minimum, should be used besides the fibroblasts.

We now show Claspin, Timeless and CHK1 protein levels in non-tumoral RPE-1, MCF10A and BJ cells (**Fig. 2b**), in addition to IMR90 (**Fig. 2a**). In all these cells, Claspin, Timeless and CHK1 levels were very low, compared to cancer cell lines (HCT116, U2OS and HeLa cells).

4. Only mRNA levels of CLSP, TIM etc. are examined in the clinical tumor material. As the protein levels often do not correlate with mRNA levels, it would be important to perform some protein analysis of a cohort of tumors - either by western blots or well-controlled immunohistochemistry, as it is the proteins that carry out the function.

A large body of published evidence indicates that Claspin and Timeless proteins are more abundant in cancer samples (see for instance Tsimaritou 2007 J Pathol 211, 331; Schepeler 2013 Oncogene 32, 3577). We have now confirmed by immunohistochemistry that Timeless is more abundant in our breast cancer cohort (**Extended data Fig. 1a**). We also provide evidence that the protein levels of Claspin, Timeless and CHK1 are largely increased in cancer cell lines (**Fig. 2a, b**) and that their levels are highly correlated in the proteomic landscape of 50 colorectal cancer cell lines (**Fig. 2c**). In this dataset, we found no correlation between Claspin/Timeless/CHK1 and other components of the ATR-CHK1 pathway, which is fully consistent with RT-qPCR in patient samples (**Extended data Fig. 1c-h**). We are therefore confident that mRNA levels in cancer samples match protein levels.

5. The experiments with the HCT116 colon cancer line are useful, however these cells are really exceptional even among tumor cell lines due to their exceptional recombination activity for which they have been used for gene knockouts in the past. It would be reassuring if at least one additional cancer cell line was used in parallel, to validate the main results and show they are representative.

We now provide evidence that the reduction of Claspin or Timeless levels affects the growth of MCF7 cells (**Extended data Fig. 3f**) to the same extent as HCT116 cells (**Fig. 3b**). We also show that the

depletion of Claspin and Timeless by siRNA increases RS in MCF7 and U2OS cells, as illustrated by spontaneous γ -H2AX levels (**Extended data Fig. 3c,d**). Importantly, this occurs again in cells that are checkpoint proficient, as for HCT116 cells (**Extended data Fig. 3c,d**).

6. Figs 2A,B, and S2A lack any comparison with normal cells.

As indicated above, we now compare protein levels in 7 cancer cell lines and immortalized normal cells (**Extended data Fig. 2a,b**). Colony formation assays were performed on HCT116 cells (**Fig. 3c**) but cannot be performed on non-transformed cells. We have put a lot of effort into depleting Claspin and Timeless with different methods to measure cell growth in immortalized fibroblasts, however, the low expression level of Claspin and Timeless does not allow us to be confident on the depletion efficiency. We therefore prefer not showing these data.

7. It is concluded that the chromatin (fork)-binding activity of claspin and timelss are increased under replication stress, however this conclusion is not sufficiently documented. What is needed is the ratios between the levels of thee proteins in supernatants and chromatin fractions of normal and cancer cells with and without hydroxyurea treatment (e.g. Fig. S1C). It looks as if the ratios are not appreciably different, only the overall levels, which is a result known for years. I.e., an alternative interpretation could be that the chromatin binding potential is NOT enhanced but is the same as in normal cells and merely parallels the overall level of the protein.

This is not exactly how we interpret the chromatin-binding experiment shown now in (**Extended data Fig. 2**). As indicated above, a recent study from the Lukas lab indicates that Timeless detaches from the replisome in response to certain types of RS (Somyajit *et al.*, Science 2017). Our data show that a large fraction of overexpressed Claspin and Timeless is found in the chromatin fraction. It is unlikely that additional Claspin and Timeless molecules are directly associated to the replisome. We rather favor the possibility that these additional proteins serve as a reservoir to promote the efficient reassembly of a functional FPC for the tolerance to RS.

8. It was unclear what was the level of phospho-Chk1 in Fig. 4C, as the panel was entirely invisible, at least in the copy I got for review.

We apologize for this technical problem occurring during the conversion to pdf.

9. There are data on the adapted clones (p. 11, end of the panultimate paragraph) quoted as not shown. This is not acceptable and such data should shown, albeit as a supplement.

As indicated in the manuscript, we have initially performed extensive analyses of differentially-expressed genes with GO-TERM, KEGG and REACTOME but these analyses failed to generate meaningful data, so there was frankly nothing much to show, even as a supplement. As discussed above, we have now performed SAM multiclass analyses focusing on different set of genes induced by E2F or E2F hyperactivation (siE2F6), HU exposure and genes involved in DNA repair. The result of these analyses is now shown in **Fig. 5f** and **Extended data Fig. 5d**.

10. fig 4F and the text at the bottom of p. 11 lack inclusion of clone 8 for consistency, and also control blots to assess the extent of siRNA mediated depletion of Tim and CLSP.

The clone #8 shows the same reduction in fork speed and low Claspin and Timeless levels as BJ-Ras cells. It is therefore not relevant to reduce Claspin or Timeless levels even more in this clone. To address the second issue, we now show Western blot of Claspin and Timeless levels transfected with siRNAs against Claspin and Timeless, or with a control siRNA (**Extended data Fig. 6a**).

11. as mentioned briefly above, the manuscript culminates by a section on NSCLC patients (p. 12), however this result, namely that levels of Tim/CLSP correlate with disease-free interval, was already published by these authors (Allera-Moreau et al. *Oncogenesis*, 2012) and therefore it is merely a confirmatory result.

See response to point 1.

12. There seems to be a discrepancy between the data in this manuscript versus those in the papers by Petermann in 2008 and the Kaufmann lab in 2007 (see refs above), in that the latter authors both observed decreased phospho-Chk1 under stress in CLSP/TIM-depleted cells, which here it is concluded that there is no change. Can the authors suggest some explanation for such discrepancy, also because the same cell line was used in some of these older studies.

This is a very important issue and we thank this Reviewer for pointing it out. As discussed now in the revised version of the manuscript, there is indeed a discrepancy in published studies (Chini 2003; Chini 2006; Liu 2006; Unsal-Kacmaz 2005; Chou, 2006) regarding the impact of Claspin/Timeless depletion on CHK1 activation. Here, we show that residual Claspin levels after depletion of 90% of the protein are still higher than those found in immortalized non-cancer cells (**Fig. 3a**) and could therefore be sufficient to activate CHK1. As proposed by others (Petermann, 2008; Scolah, 2009), Claspin and Timeless could be functionally redundant with other adaptors (BRCA1, TopBP1), which are co-regulated with the downstream components of the ATR-CHK1 pathway at the protein level (**Fig. 2c**). Nevertheless, the point of our study was not to re-investigate the checkpoint function of Claspin and Timeless but to show that a level of depletion that does not affect ATR-CHK1 signaling strongly affects replication forks. Despite the lack of *bona fide* separation-of-function mutants, this allows us to conclude that the fork protection function of Claspin and Timeless promotes tolerance to oncogene-induced RS in cancer cells, and not the checkpoint function.

Reviewer #2 (Remarks to the Author):

This study by Bianco et al. reports that selected primary cancers and cancer cells specifically overexpress the replication stress response factors Claspin, Timeless and Chk1, but not other central components of the replication stress response such as ATR. This suggests that elevated levels of the former proteins are important for the proliferation of cancer cells. The study then concentrates on Claspin and Timeless and shows that these promote the proliferation of cancer cells and this is independent of ATR signalling through Chk1 and Cdc25A. Interestingly, oncogenic Ras-expressing fibroblasts that have escaped senescence also have increased levels of Claspin, Chk1 and Timeless, suggesting that these proteins can allow cells to proliferate in presence of oncogene-induced replication stress. Finally the authors suggest that overexpression of these proteins correlates with poor outcomes in cancer patients and that they could pose potential targets for cancer therapy.

The data are of good quality and provide new insights are potentially of interest to the wider field. Particular strengths are the in-house analysis of tumour using qRT-PCR (rather than just high-throughput data) in Figure 1 and the generation and analysis of Ras-induced senescence resistant clones in Figure 4.

However it has been reported previously that depletion of Claspin, Chk1 or Timeless reduces replication fork progression in cancer cells, including HCT116 cells (refs 43,44 cited by the authors and Unsal-Kacmaz K, Mol Cell Biol 2007). The independence of Claspin function from Chk1-Cdc25A has also been reported (ref 44). The novelty of this study therefore rests on whether it can make a convincing case that maintenance of replication fork progression by these proteins is more important for proliferation of cancer cells than normal cells. There are some questions in whether this has been achieved. Further evidence will be required to strengthen this conclusion.

We thank this Reviewer for his/her interest in our study and for his/her constructive comments. As discussed in the answer to Reviewer #1, the novelty of our study does not lie in another analysis of the dual functions of Claspin and Timeless in checkpoint signaling and fork protection but in the determination of which of these two functions are important to tolerate oncogene-induced replication stress in cancer cells. We apologize if this was not clear enough in the previous version of our manuscript and we hope that it is now better explained in this revised version.

Main points:

- Is there evidence that the increased Chk1, Claspin and TIM mRNA levels in primary lung, breast and colorectal cancers translate into increased protein levels (Figure 1)?

As indicated in our response to point #4 of Reviewer #1, a large body of published evidence indicates that Claspin and Timeless proteins are more abundant in cancer samples (see for instance Tsimaratou 2007 J Pathol 211, 331; Schepeler 2013 Oncogene 32, 3577). We have now confirmed by immunohistochemistry that Timeless is more abundant in our breast cancer cohort (**Extended data Fig. 1a**). We also provide evidence that the protein levels of Claspin, Timeless and CHK1 are largely increased in cancer cell lines (**Fig. 2a, b**) and that their levels are highly correlated in the proteomic landscape of 50 colorectal cancer cell lines (**Fig. 2c**). In this dataset, we found no correlation between Claspin/Timeless/CHK1 and other components of the ATR-CHK1 pathway, which is fully consistent with RT-qPCR in patient samples (**Extended data Fig. 1c-h**). We are therefore confident that mRNA levels in cancer samples match protein levels.

- Are Claspin and Timeless only required for the proliferation of HCT116 cancer cells or would the same effects happen in normal cells (Figure 2, Figure 3)? There text argues that HCT116 cells still

have more Claspin and Timeless after shRNA depletion than untreated IMR90 cells, yet their growth is still impaired. But has this been normalised for S phase content?

Claspin and Timeless are essential proteins and Timeless KO is embryonic lethal in mice. A complete depletion in IMR90 cells would therefore be deleterious. Yet, we have tried to perform the experiment suggested by this Reviewer and repeatedly obtained heterogenous results, with a large fraction of cells losing viability and a small population resisting to transfection that would eventually overgrow the others. As for the normalization to S-phase cells, we initially presented data showing that 17% of IMR90 cells, compared to 37 to 49% in HeLa, HCT116 and U2OS cells (previous Fig S1B). The aim of this analysis was to normalize the amount of Claspin and Timeless in cancer cell lines and normal cells to the fraction of cells in S phase. However, since the amount of Claspin in IMR90 cells was barely detectable, we could not perform an accurate normalization and we decided to remove these flow cytometry profiles.

- Figure S1 shows that HCT116, HeLa and U2OS cells have larger S phase population than IMR90 cells. It is mentioned in the text that protein levels have been normalised to S phase fraction, but these data should be shown. Normalisation of other signals such as gamma-H2AX should also be shown.

See above answer, but we are happy to reintegrate this analysis if this Reviewer consider it important.

- If increased fork speeds due to Claspin and TIM overexpression allow cells to adapt to Ras-induced replication stress, then overexpressing these proteins in BJ-Ras cells should protect those from senescence (Figure 4). Can this be tested?

This experiment is technically very challenging as CLSPN and TIMELESS are very large genes and BJ cells are difficult to transfect. Since cells need to be kept in culture 6 to 10 days after transfection to monitor senescence, it is difficult to maintain a constant level of expression in the population of cells. Yet, we were able to detect a partial rescue of the slow fork phenotype induced by Ras in these cells (**Extended data Fig. 6c**). We have also confirmed this result in U2OS cells overexpressing CycE (**Extended data Fig. 6d**). Since senescence is directly caused by RS in BJ cells, it is therefore likely that the overexpression of Claspin/Timeless would protect BJ-Ras cells from senescence.

- What is the effect of Claspin and TIM depletion on growth of clones 4, 5 and 8 (Figure 4)? What is the effect on DNA damage signals such as gamma-H2AX? It would be much more helpful to have the analyses of Figure 2 and 3 done in the adapted clones versus BJ and BJ-Ras cells.

We show now the effect of Claspin and Timeless depletion on gamma-H2AX activation (**Extended data Fig. 6a**) and fork progression (**Fig. 6b**) in those BJ-Ras clones.

- Page 6: “The overexpression of CLSPN and CHK1 correlates with expression cell proliferation marker Ki67 in lung adenocarcinomas, but not in breast and colon cancers (data not shown).” Please show these data. Secondly, if CLSPN and CHK1 overexpression correlate with proliferation in lung cancers, then how informative is the effect of this overexpression on disease-free survival in Figure 5?

We now provide evidence that the levels of Claspin, Timeless and CHK1 mRNAs correlate with PCNA in lung cancer, but not in colorectal and breast cancers (**Fig. 1c**). Yet, it should be noticed that the level of PCNA mRNA in lung cancer increases only modestly (1.4 fold) compared to those of Claspin and CHK1 (4.5- and 4.4-fold, respectively; **Fig. 1b**). It is therefore unlikely that the enhanced expression of Claspin, Timeless and CHK1 in cancer cells simply reflects increased proliferation.

Other comments:

Page 9: “These data are consistent with results obtained in other cancer cell lines 43,44.” Work in ref 43 was in fact performed in HCT116 cell as well. Secondly, Unsal-Kacmaz K, Mol Cell Biol 2007, previously reported fork slowing induced by Timeless depletion and should be cited.

Thanks, we have now corrected this statement and added the missing reference, which was also cited elsewhere.

Figure 4C: The phospho-Chk1 blot is not visible in my version of the PDF

Figure S1C: Loading control for chromatin should be added

Figure S2C: The phospho-Chk1 blot is not visible

We apologize for these technical problems due to file conversion. We now show histone H3 as loading control for the chromatin fraction.

BJ-hTert cells should not be referred to as primary cells; they are immortalised non-cancer cells, which is not the same.

Indeed, this has now been corrected in the manuscript.

Reviewer #3 (Remarks to the Author):

This manuscript proposes that cancer cells adapt to oncogene-induced replication stress by overexpressing claspin and timeless.

My opinion is that the main premise of this manuscript is correct and important. Indeed, in a number of settings, timeless, tipin and claspin have been shown to be important for the response of cells to DNA replication stress. My main concern is novelty, as similar observations have been reported previously. Of course, this manuscript goes beyond what has been previously published, but by the way the manuscript is written, this is not clear. Previous observations made by others should be cited in the Introduction and not in the Discussion. A full list of relevant papers should be cited. And the questions addressed by this manuscript and the advances made should be clearly stated.

We are grateful to this Reviewer for his/her suggestions. We have followed this advice and completely reorganized the introduction and discussion to stress the novelty of our work.

Specific Comments

1. Others have previously reported that timeless and claspin are overexpressed in human cancer. The authors cite a few papers in the Discussion, but from the Introduction, it appears that nobody has looked at this before. This should be fixed. The authors should also cite a full list of relevant papers. I quickly found:

- Yoshida, Sato et al, Cancer Science 2013, showing overexpression of timeless in lung cancer and poor correlation with patient survival (not cited by the authors)
- Fu, Leaderer et al., Mol Carcinog 2011, overexpression of timeless in breast cancer and link to risk (not cited)
- Baldeyron et al., Mol Oncol 2015, tipin overexpression in cancer (not cited).

The authors have, however, cited several relevant papers on this point in the Discussion.

Thanks for pointing this out. We did not cite Baldeyron *et al.* on purpose, precisely because in concerns Tipin and not Timeless, but it is now cited, together with the other references.

2. In Fig 3A, it is difficult to position the gate to identify the gH2AX-positive cells, because the difference between the gH2AX negative and positive signals is small. How many times was this experiment repeated? Perhaps, gH2AX by FACS is not the best assay for replication stress in this system.

The FACS profiles with a control antibody used to position the gate is now shown in **Extended data Fig. 4a**. The experiment was performed twice. We agree on the fact that flow cytometry is not the most sensitive approach to quantify γ -H2AX but the point here was to verify that the signal detected on western blots is specific to S-phase cells. A similar result was obtained after cell sorting (**Fig. 4f**) and by quantitative immunofluorescence (**Fig. 4a,b**).

3. Fig 3 C and F, depletion of claspin reduces CldU track length in both early and late S phase cells, but depletion of timeless reduces CldU track length only in early S. What is the explanation for this different behavior? How many times has this experiment been done?

The depletion of Timeless actually reduces track length only in late S phase (**Fig. 4g**). This illustrates a functional difference between Claspin and Timeless that is now extensively discussed in the Discussion section of the manuscript. All these experiments were done in triplicate.

4. Fig. 4A. How many times has this experiment been performed? There are no error bars.

The selection of the clones is a very long process that was done only once. Several technical replicates have been performed for the quantification of mRNAs and one representative experiment is shown.

5. Have all the experiments with cell lines been performed in triplicate?

All experiments with cell lines were performed at least three times, except for the γ -H2AX immunofluorescence (twice, **Fig. 4a**) and CNV analysis (once, **Extended data Fig. 4e,f**).

REVIEWERS' COMMENTS:

Reviewer #1 (Remarks to the Author):

The authors have addressed most of my concerns adequately, however the main issue of providing mechanistic insight into how timeless and claspin protect replication forks in a checkpoint-independent manner has remained unclear. The concerns related to insufficient controls have been resolved, and generally the technical side of the study is now acceptable. While this is a borderline case in terms of whether the paper provides sufficient advance to warrant publication in Nat Comms, I am inclined to support the acceptance of the revised manuscript, in case the other referees share this improved opinion.

Reviewer #2 (Remarks to the Author):

The authors have addressed several of my concerns and I now consider this manuscript acceptable for publication.

Reviewer #3 (Remarks to the Author):

In this revised version of the manuscript, the authors have addressed my concerns. Specifically, they more clearly state the novelty of the manuscript and explain that Claspin and Timeless have 2 functions (fork protection and checkpoint activation) and that only the first function promotes tolerance to replication stress in cancer cells.

Overall, I support publication of this revised version of the manuscript. Two minor points that the authors could address:

1. I wonder if the title could be made more specific and better describe the novel message of this manuscript.
2. The abstract needs a little more work. The sentence: "Here, we were able to separate these two functions (fork protection and checkpoint signaling) by reducing Claspin or Timeless expression to a level that increases endogenous RS without affecting checkpoint signaling" is not followed by a sentence saying that the first function (fork protection) promotes tolerance to RS.

Reviewer #1 (Remarks to the Author):

The authors have addressed most of my concerns adequately, however the main issue of providing mechanistic insight into how timeless and claspin protect replication forks in a checkpoint-independent manner has remained unclear. The concerns related to insufficient controls have been resolved, and generally the technical side of the study is now acceptable. While this is a borderline case in terms of whether the paper provides sufficient advance to warrant publication in Nat Comms, I am inclined to support the acceptance of the revised manuscript, in case the other referees share this improved opinion.

Reviewer #2 (Remarks to the Author):

The authors have addressed several of my concerns and I now consider this manuscript acceptable for publication.

Reviewer #3 (Remarks to the Author):

In this revised version of the manuscript, the authors have addressed my concerns. Specifically, they more clearly state the novelty of the manuscript and explain that Claspin and Timeless have 2 functions (fork protection and checkpoint activation) and that only the first function promotes tolerance to replication stress in cancer cells.

We thank all three Reviewers for acknowledging the fact that we have significantly improved our manuscript and that it is now suitable for publication. We also thank them for their constructive comments and for helping us improve this manuscript.

Overall, I support publication of this revised version of the manuscript. Two minor points that the authors could address:

1. I wonder if the title could be made more specific and better describe the novel message of this manuscript.

We have modified the title to “Overexpression of Claspin and Timeless protects cancer cells from replication stress in a checkpoint-independent manner”.

2. The abstract needs a little more work. The sentence: "Here, we were able to separate these two functions (fork protection and checkpoint signaling) by reducing Claspin or Timeless expression to a level that increases endogenous RS without affecting checkpoint signaling" is not followed by a sentence saying that the first function (fork protection) promotes tolerance to RS.

We have extensively modified the abstract to shorten it to less than 150 words and to address the issue raised by this Reviewer.